# Reward-Free Curricula for Training Robust World Models

**Marc Rigter**
University of Oxford
`marcrigter@gmail.com`

**Minqi Jiang**
University College London

**Ingmar Posner**
University of Oxford

## Abstract

There has been a recent surge of interest in developing *generally-capable agents* that can adapt to new tasks without additional training in the environment. Learning *world models* from *reward-free* exploration is a promising approach, and enables policies to be trained using imagined experience for new tasks. However, achieving a general agent requires *robustness* across different environments. In this work, we address the novel problem of generating curricula in the reward-free setting to train robust world models. We consider robustness in terms of *minimax regret* over all environment instantiations and show that the minimax regret can be connected to minimising the maximum error in the world model across environment instances. This result informs our algorithm, *WAKER: Weighted Acquisition of Knowledge across Environments for Robustness*. WAKER selects environments for data collection based on the estimated error of the world model for each environment. Our experiments demonstrate that WAKER outperforms several baselines, resulting in improved robustness, efficiency, and generalisation.

## 1 Introduction

Deep reinforcement learning (RL) has been successful on a number of challenging domains, such as Go (Silver et al., 2016), and Starcraft (Vinyals et al., 2019). While these domains are difficult, they are narrow in the sense that they require solving a single pre-specified task. More recently, there has been a surge of interest in developing *generally-capable agents* that can master many tasks and quickly adapt to new tasks without any additional training in the environment (Brohan et al., 2022; Mendonca et al., 2021; Reed et al., 2022; Sekar et al., 2020; Stooke et al., 2021; Jiang et al., 2022b).

Motivated by the goal of developing generalist agents that are not specialised for a single task, we consider the *reward-free* setting. In this setting, the agent first accumulates useful information about the environment in an initial, reward-free exploration phase. Afterwards, the agent is presented with specific tasks corresponding to arbitrary reward functions, and must quickly adapt to these tasks by utilising the information previously acquired during the reward-free exploration phase. By separating the learning of useful environment representations into an initial pre-training phase, reward-free learning provides a powerful strategy for data-efficient RL.

A promising line of work in the reward-free setting involves learning *world models* (Ha & Schmidhuber, 2018), a form of model-based RL (Sutton, 1991), where the agent learns a predictive model of the environment. In the reward-free setting, the world model is trained without access to a reward function, and instead, is trained using data collected by a suitable exploration policy (Sekar et al., 2020). Once a world model has been trained for an environment, it is possible to train policies entirely within the world model (i.e. "in imagination") for new tasks corresponding to specific reward functions within that environment (Sekar et al., 2020; Xu et al., 2022; Rajeswar et al., 2023).

However, to realise the vision of a general agent, it is not only necessary for the agent to be able to learn multiple tasks in a single environment: the agent must also be robust to different environments. To enable this, one approach is to apply *domain randomisation* (DR) (Tobin et al., 2017) to sample different environments uniformly at random to gather a more diverse dataset. However, the amount of data required to learn a suitable world model might vary by environment. *Unsupervised Environment Design* (UED) (Dennis et al., 2020) aims to generate curricula that present the optimal environments to the agent at each point of training, with the goal of maximising the robustness of

the final agent across a wide range of environments. However, existing UED approaches require a task-specific reward function during exploration (Eimer et al., 2021; Matiisen et al., 2019; Portelas et al., 2020; Dennis et al., 2020; Jiang et al., 2021a; Mehta et al., 2020; Parker-Holder et al., 2022; Wang et al., 2019), and therefore cannot be applied in the reward-free setting that we consider. In this work we address the novel problem of generating curricula for training robust agents *without access to a reward function* during exploration. To distil the knowledge obtained during reward-free exploration, we aim to learn a world model that is robust to downstream tasks and environments.

We first analyse the problem of learning a robust world model in the reward-free setting. We then operationalise these insights in the form of novel algorithms for robust, reward-free world model learning. Inspired by past works on UED with known reward functions (Dennis et al., 2020; Jiang et al., 2021a; Parker-Holder et al., 2022), we consider robustness in terms of minimax regret (Savage, 1951). To our knowledge, WAKER is the *first work to address automatic curriculum learning for environment selection without access to a reward function*. We make the following contributions: a) We formally define the problem of learning a robust world model in the reward-free setting, in terms of minimax regret optimality, b) We extend existing theoretical results for MDPs to prove that this problem is equivalent to minimising the maximum expected error of the world model across all environments under a suitable exploration policy, and finally c) We introduce WAKER, an algorithm for actively sampling environments for exploration during reward-free training based on the estimated error of the world model in each environment. We introduce pixel-based continuous control domains for benchmarking generalisation in the reward-free setting. We evaluate WAKER on these domains, by pairing it with both an instrinsically-motivated exploration policy and a random exploration policy. Our results show that WAKER outperforms several baselines, producing more performant and robust policies that generalise better to out-of-distribution (OOD) environments.

## 2 PRELIMINARIES

A *reward-free* Markov Decision Process (MDP) is defined by $\mathcal{M} = \{S, A, T\}$, where $S$ is the set of states and $A$ is the set of actions. $T : S \times A \to \Delta(S)$ is the transition function, where $\Delta(S)$ denotes the set of possible distributions over $S$. For some reward function, $R : S \times A \to [0, 1]$, we write $\mathcal{M}^R$ to denote the corresponding (standard) MDP (Puterman, 2014) with reward function $R$ and discount factor $\gamma$. A reward-free Partially Observable Markov Decision Process (POMDP) (Kaelbling et al., 1998) is defined by $\mathcal{P} = \{S, A, O, T, I\}$, where $O$ is the set of observations, and $I : S \to \Delta(O)$ is the observation function. A *history* is a sequence of observations and actions, $h = o_0, a_0, \ldots, o_t, a_t$, where $o_i \in O$ and $a_i \in A$. We use $\mathcal{H}$ to denote the set of all possible histories. Analogous to the MDP case, $\mathcal{P}^R$ denotes a POMDP with reward function $R$ and discount factor $\gamma$.

We assume that there are many possible instantiations of the environment. To model an underspecified environment, we consider a reward-free Underspecified POMDP (UPOMDP): $\mathcal{U} = \{S, A, O, T_\Theta, I, \Theta\}$ (Dennis et al., 2020). In contrast to a POMDP, the UPOMDP additionally includes a set of free parameters of the environment, $\Theta$. Furthermore, the transition function depends on the setting of the environment parameters, i.e. $T_\Theta : S \times A \times \Theta \to \Delta(S)$. For each episode, the environment parameters are set to a specific value $\theta \in \Theta$. Therefore, for each episode the environment can be represented by a standard POMDP $\mathcal{P}_\theta = \{S, A, O, T_\theta, I, \gamma\}$, where $T_\theta(s, a) = T_\Theta(s, a, \theta)$.

**World Models** Model-based RL algorithms use experience gathered by an agent to learn a model of the environment (Sutton, 1991; Janner et al., 2019). When the observations are high-dimensional, it is beneficial to learn a compact latent representation of the state, and predict the environment dynamics in this latent space. Furthermore, in partially observable environments where the optimal action depends on the history of observations and actions, recurrent neural networks can be used to encode the history into a Markovian representation (Schmidhuber, 1990; Karl et al., 2017). In this work, we consider a *world model* to be a model that utilises a recurrent module to predict environment dynamics in a Markovian latent space (Ha & Schmidhuber, 2018; Hafner et al., 2021).

Let the environment be some reward-free POMDP, $\mathcal{P}$. A world model, $W$, can be thought of as consisting of two parts: $W = \{q, \widehat{T}\}$. The first part is the representation model $q : \mathcal{H} \to Z$, which encodes the history into a compact Markovian latent representation $z \in Z$. The second part is the latent transition dynamics model, $\widehat{T} : Z \times A \to \Delta(Z)$, which predicts the dynamics in latent space. Because the latent dynamics are Markovian, we can think of the world model is approximating the

original reward-free POMDP $\mathcal{P}$ with a reward-free MDP in latent space: $\widehat{\mathcal{M}} = (Z, A, \widehat{T})$. In this work, we consider policies of the form: $\pi : Z \rightarrow \Delta(A)$. This corresponds to policies that are Markovian in latent space, and history-dependent in the original environment.

**Minimax Regret** In *robust* optimisation, there is a set of possible scenarios, each defined by parameters $\theta \in \Theta$. The goal is to find a solution that achieves strong performance across all scenarios. In the context of reinforcement learning, we can think of each scenario as a possible instantiation of the environment, $\mathcal{P}_\theta$. For some reward function, $R$, the expected value of a policy in $\mathcal{P}_\theta^R$ is $V(\pi, \mathcal{P}_\theta^R) := \mathbb{E}[\sum_{t=0}^\infty \gamma^t r_t \mid \pi, \mathcal{P}_\theta^R]$, where $r_t$ are the rewards received by executing $\pi$ in $\mathcal{P}_\theta^R$.

Minimax regret (Savage, 1951) is a commonly used objective for robust policy optimisation in RL (Chen et al., 2022; Dennis et al., 2020; Jiang et al., 2021a; Parker-Holder et al., 2022; Rigter et al., 2021). To define the minimax regret objective, we begin by defining the optimal policy for a given environment and reward function, $\pi_{\theta,R}^* = \arg\max_\pi V(\pi, \mathcal{P}_\theta^R)$. We refer to each $\mathcal{P}_\theta^R$ as a "task" or, when clear from context, an "environment." The regret for some arbitrary policy $\pi$ in environment $\mathcal{P}_\theta^R$ is

$$\text{REGRET}(\pi, \mathcal{P}_\theta^R) := V(\pi_{\theta,R}^*, \mathcal{P}_\theta^R) - V(\pi, \mathcal{P}_\theta^R). \tag{1}$$

The minimax regret objective finds the policy with the lowest regret across *all* environments:

$$\pi_{\text{regret}}^* = \arg\min_\pi \max_{\theta \in \Theta} \text{REGRET}(\pi, \mathcal{P}_\theta^R). \tag{2}$$

Minimax regret aims to find a policy that is *near-optimal* in all environments, and is therefore robust.

## 3 APPROACH

The minimax regret objective defines how to optimise a *policy* to be robust to different environments *when the task is known*. However, our aim in this work is to train a world model such that policies *derived from the world model* for *future tasks* are robust to different environments. In this section, we present our approach for gathering data to train a robust world model to achieve this aim.

In Section 3.1, we outline how we learn a single world model for many possible environments. In Section 3.2 we define the Reward-Free Minimax Regret objective, which connects minimax regret to world model training by assuming that when a reward function is provided, the optimal policy in the world model for that reward function can be found. We then show that we can optimise an upper bound on this objective by minimising the maximum expected latent dynamics error in the world model across all environments, under a suitable exploration policy. This informs our algorithm for selecting environments to sample data from to train the world model, *WAKER: Weighted Acquisition of Knowledge across Environments for Robustness* (Section 3.4). WAKER biases sampling towards environments where the world model is predicted to have the greatest errors (Figure 1a).

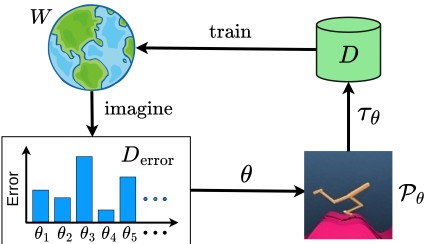

(a) WAKER overview.

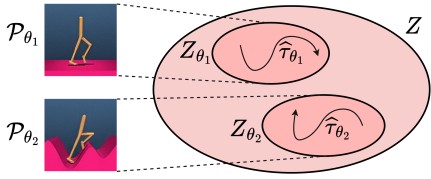

(b) Illustration of world model latent space for a UPOMDP.

Figure 1: a) WAKER uses error estimates for each environment to choose the next environment to sample data from, $\mathcal{P}_\theta$. A trajectory $\tau_\theta$ is collected by rolling out exploration policy $\pi^{\text{expl}}$ in the selected environment. $\tau_\theta$ is added to the data buffer $D$ which is used to train the world model, $W$. Imagined trajectories in $W$ are used to update the error estimates. b) In the world model, each environment is encoded to a subset, $Z_\theta$, of the latent space $Z$ by representation model $q$.

### 3.1 WORLD MODELS FOR UNDERSPECIFIED POMDPS

We utilise a single world model, $W = \{q, \widehat{T}\}$, to model the reward-free UPOMDP. Consider any parameter setting of the UPOMDP, $\theta \in \Theta$, and the corresponding reward-free POMDP, $\mathcal{P}_\theta$. For any history in $\mathcal{P}_\theta$, $h \in \mathcal{H}_\theta$, the representation model encodes this into a subset of the latent space, i.e. $q : \mathcal{H}_\theta \rightarrow Z_\theta$, where $Z_\theta \subset Z$. We can then use the latent dynamics model $\widehat{T}$ to predict the

latent dynamics in $Z_\theta$, corresponding to the dynamics of $\mathcal{P}_\theta$ for any $\theta \in \Theta$. Thus, we think of the world model as representing the set of reward-free POMDPs in the UPOMDP by a set of reward-free MDPs, each with their own latent state space: $\widehat{\mathcal{M}}_\theta = \{Z_\theta, A, \widehat{T}, \gamma\}$. This is illustrated in Figure 1b.

Using a single world model across all environment parameter settings is a natural approach as it a) utilises the recurrent module to infer the parameter setting (which may be partially observable), and b) enables generalisation between similar parameter settings. Furthermore, it is sufficient to train a *single generalist policy* over the entire latent space $Z$ to obtain a policy for all environments.

### 3.2 Reward-Free Minimax Regret: Problem Definition

As discussed in Section 2, we can define robust policy optimisation via the minimax regret objective. However, this definition cannot be directly applied to our setting where we do not know the reward function during exploration, and our goal is to train a world model. In this section, we present the first contribution of this work, the Reward-Free Minimax Regret problem, which adapts the minimax regret objective to the setting of reward-free world model training that we address.

Consider some world model, $W$, which as discussed in Section 3.1 represents the possible environments by a set of reward-free MDPs in latent space, $\{\widehat{\mathcal{M}}_\theta\}_{\theta \in \Theta}$. Assume that after training $W$ we are given some reward function, $R$. We define $\widehat{\pi}_{\theta,R}^*$ to be the *optimal world model policy* for that reward function $R$ and parameter setting $\theta$, i.e.

$$\widehat{\pi}_{\theta,R}^* = \arg \max_\pi V(\pi, \widehat{\mathcal{M}}_\theta^R). \tag{3}$$

This is the optimal policy according to the MDP defined in the latent space of the world model for parameter setting $\theta$ and reward function $R$, and does not necessarily correspond to the optimal policy in the real environment. From here onwards, we will make the following assumption.

**Assumption 1** *Consider some world model $W$ that defines a set of MDPs in latent space $\{\widehat{\mathcal{M}}_\theta\}_{\theta \in \Theta}$. Assume that given any reward function $R$, and parameter setting $\theta \in \Theta$, we can find $\widehat{\pi}_{\theta,R}^*$.*

Assumption 1 is reasonable because we can generate unlimited synthetic training data in the world model for any parameter setting, and we can use this data to find a policy that is near-optimal in the world model using RL. In practice, we cannot expect to find the exact optimal policy within the world model, however Assumption 1 enables an analysis of our problem setting. We now define the Reward-Free Minimax Regret problem.

**Problem 1 (Reward-Free Minimax Regret)** *Consider some UPOMDP, $\mathcal{U}$, with parameter set $\Theta$. For world model $W$, and $\theta \in \Theta$, let $\widehat{\mathcal{M}}_\theta^R$ be the latent-space MDP defined by $W$, that represents real environment $\mathcal{P}_\theta$ with reward function $R$. Define the optimal world model policy as $\widehat{\pi}_{\theta,R}^* = \arg \max_\pi V(\pi, \widehat{\mathcal{M}}_\theta^R)$. Find the world model, $W^*$, that minimises the maximum regret of the optimal world model policy across all parameter settings and reward functions:*

$$W^* = \arg \min_W \max_{\theta,R} \text{REGRET}(\widehat{\pi}_{\theta,R}^*, \mathcal{P}_\theta^R) \tag{4}$$

Problem 1 differs from the standard minimax regret objective in Equation 2 in two ways. First, Problem 1 optimises a world model (not a policy) under the assumption that the optimal world model policy can later be recovered (i.e. Assumption 1). Second, the maximum is over all possible reward functions in addition to parameter settings. This makes Problem 1 suitable for the reward-free setting: we do not need access to a specific reward function when training the world model.

### 3.3 Theoretical Motivation

Before presenting our algorithm for Problem 1, we first provide the motivation for our approach. We make the assumption that the world model learns a suitable representation model, $q$, which encodes any sequence of observations and actions in the UPOMDP into a Markovian latent state.

**Assumption 2** *Consider the representation model learnt by the world model, $q : \mathcal{H}_\theta \rightarrow Z_\theta$, for all $\theta \in \Theta$. Assume that given the representation model $q$, there exists a latent transition dynamics function, $T$, for which the expected reward is the same as the real environment: $V(\pi, \mathcal{P}_\theta^R) = V(\pi, \mathcal{M}_\theta^R)$, where $\mathcal{M}_\theta^R = (Z_\theta, A, R, T, \gamma)$, for any policy $\pi$, reward function $R$, and parameter setting $\theta$.*

Assumption 2 states that the representation model successfully encodes any sequence of observations and actions into a Markovian latent state. Therefore, there exists a dynamics function $T$ defined over the latent space that exactly models the real environment. Assumption 2 allows us to reason about the inaccuracy of the world model solely in terms of the difference between the learnt latent dynamics function, $\widehat{T}$, and the (unknown) exact latent dynamics, $T$. For this reason, from here onwards we will *solely refer to the latent dynamics function* $\widehat{T}$ when discussing the world model, under the implicit assumption that a suitable representation model $q$ is learnt according to Assumption 2.

Using Assumption 2 we can bound the sub-optimality of the optimal world model policy for any parameter setting $\theta$ according the difference between the learnt latent dynamics function, $\widehat{T}$, and the true latent dynamics, $T$, in latent MDP $\widehat{\mathcal{M}}_\theta$. This is stated formally in Proposition 1.

**Proposition 1** *Let $\widehat{T}$ be the learnt latent dynamics in the world model. Assume the existence of a representation model q that adheres to Assumption 2, and let T be the true latent dynamics according to Assumption 2. Then, for any parameter setting $\theta$ and reward function R, the regret of the optimal world model policy is bounded according to:*

$$\text{REGRET}(\widehat{\pi}_{\theta,R}^*, \mathcal{P}_\theta^R) \leq \frac{2\gamma}{(1-\gamma)^2}\Big[\mathbb{E}_{z,a\sim d(\pi_{\theta,R}^*,\widehat{\mathcal{M}}_\theta)}\big[\text{TV}\big(\widehat{T}(\cdot|z,a), T(\cdot|z,a)\big)\big]$$
$$+ \mathbb{E}_{z,a\sim d(\widehat{\pi}_{\theta,R}^*,\widehat{\mathcal{M}}_\theta)}\big[\text{TV}\big(\widehat{T}(\cdot|z,a), T(\cdot|z,a)\big)\big]\Big]$$

*where $d(\pi, \mathcal{M})$ denotes the state-action distribution of $\pi$ in MDP $\mathcal{M}$, and $\text{TV}(P, Q)$ is the total variation distance between distributions $P$ and $Q$.*

*Proof Sketch*: This can be proven by applying a version of the Simulation Lemma (Kearns & Singh, 2002) twice. The full proof is provided in Appendix B.

Proposition 1 tells us that the optimal world model policy will have low regret if $\widehat{T}$ is accurate under the latent state-action distribution of both $\pi_{\theta,R}^*$ and $\widehat{\pi}_{\theta,R}^*$ in $\widehat{\mathcal{M}}_\theta$. However, during data collection we do not have access to the reward function. Therefore, we do not know these distributions as the state-action distribution induced by both $\pi_{\theta,R}^*$, and $\widehat{\pi}_{\theta,R}^*$ depends upon the reward function. To alleviate this issue, we define an exploration policy, $\pi_\theta^{\text{expl}}$, that maximises the expected error (in terms of total variation distance) of the latent dynamics model:

$$\pi_\theta^{\text{expl}} = \arg\max_\pi \mathbb{E}_{z,a\sim d(\pi,\widehat{\mathcal{M}}_\theta)}\Big[\text{TV}\big(\widehat{T}(\cdot|z,a), T(\cdot|z,a)\big)\Big]. \tag{5}$$

This allows us to write an upper bound on the regret that has no dependence on the reward function:

$$\text{REGRET}(\widehat{\pi}_{\theta,R}^*, \mathcal{P}_\theta^R) \leq \frac{4\gamma}{(1-\gamma)^2}\mathbb{E}_{z,a\sim d(\pi_\theta^{\text{expl}},\widehat{\mathcal{M}}_\theta)}\Big[\text{TV}\big(\widehat{T}(\cdot|z,a), T(\cdot|z,a)\big)\Big] \text{ for all } R. \tag{6}$$

Therefore, we can upper bound the objective of Problem 1 by the maximum expected TV error in the latent dynamics function over all parameter settings:

$$\max_{\theta,R} \text{REGRET}(\widehat{\pi}_{\theta,R}^*, \mathcal{P}_\theta^R) \leq \max_\theta \frac{4\gamma}{(1-\gamma)^2}\mathbb{E}_{z,a\sim d(\pi_\theta^{\text{expl}},\widehat{\mathcal{M}}_\theta)}\Big[\text{TV}\big(\widehat{T}(\cdot|z,a), T(\cdot|z,a)\big)\Big]. \tag{7}$$

We now formally state the Minimax World Model Error problem, which proposes to optimise this upper bound as an approximation to the Reward-Free Minimax Regret objective in Problem 1.

**Problem 2 (Minimax World Model Error)** *Consider some UPOMDP, $\mathcal{U}$, with parameter set $\Theta$, and world model latent dynamics function $\widehat{T}$. Let T be the true latent dynamics function according to Assumption 2. Define the world model error for some parameter setting $\theta$ as:*

$$\text{WORLDMODELERROR}(\widehat{T}, \theta) = \mathbb{E}_{z,a\sim d(\pi_\theta^{\text{expl}},\widehat{\mathcal{M}}_\theta)}\Big[\text{TV}\big(\widehat{T}(\cdot|z,a), T(\cdot|z,a)\big)\Big] \tag{8}$$

*Find the world model that minimises the maximum world model error across all parameter settings:*

$$T^* = \arg\min_{\widehat{T}} \max_{\theta\in\Theta} \text{WORLDMODELERROR}(\widehat{T}, \theta) \tag{9}$$

Problem 2 optimises an upper bound on our original objective in Problem 1 (see Equation 7). Problem 2 finds a world model that has low prediction error across all environments under an exploration

policy that seeks out the maximum error. This ensures that for any future reward function, the optimal world model policy will be near-optimal for all environments. In the next section, we present our algorithm for selecting environments to gather data for world model training, with the aim of minimising the maximum world model error across all environments, per Problem 2.

### 3.4 WEIGHTED ACQUISITION OF KNOWLEDGE ACROSS ENVIRONMENTS FOR ROBUSTNESS

**Overview** In this section, we address how to select environments to collect data for world model training, so that the world model approximately solves Problem 2. We cannot directly evaluate the world model error in Equation 8, as it requires knowing the true latent dynamics function, $T$. Therefore, following works on offline RL (Lu et al., 2022; Yu et al., 2020; Kidambi et al., 2020), we use the disagreement between an ensemble of neural networks as an estimate of the total variation distance between the learnt and true latent transition dynamics functions in Equation 8. This enables us to generate an estimate of the world model error for each environment. Then, when sampling environments, our algorithm (WAKER) biases sampling towards the environments that are estimated to have the highest error. By gathering more data for the environments with the highest estimated error, WAKER improves the world model on those environments, and therefore WAKER reduces the maximum world model error across environments as required by Problem 2.

**WAKER** is presented in Algorithm 1, and illustrated in Figure 1a. We train a single exploration policy over the entire latent space to approximately optimise the exploration objective in Equation 5 across all environments. Therefore, we refer to the exploration policy simply as $\pi^{\text{expl}}$, dropping the dependence on $\theta$. We maintain a buffer $D_{\text{error}}$ of the error estimate for each parameter setting $\theta$ that we have collected data for. To choose the environment for the next episode of exploration, with probability $p_{\text{DR}}$ we sample $\theta$ using domain randomisation (Line 5). This ensures that we will eventually sample all environments. With probability $1 - p_{\text{DR}}$, we sample $\theta$ from a Boltzmann distribution, where the input to the Boltzmann distribution is the error estimate for each environment in $D_{\text{error}}$ (Line 7). Once the environment has been selected we sample a trajectory, $\tau_\theta$, by rolling out $\pi^{\text{expl}}$ in the environment (Line 8). We add $\tau_\theta$ to the data buffer $D$, and perform supervised learning on $D$ (Line 10) to update the world model.

---

**Algorithm 1** Weighted Acquisition of Knowledge across Environments for Robustness (WAKER)

1: **Inputs:** UPOMDP with free parameters $\Theta$; Boltzmann temperature $\eta$; Imagination horizon $h$;
2: **Initialise:** data buffer $D$; error buffer $D_{\text{error}}$; world model $W = \{q, \widehat{T}\}$; exploration policy $\pi^{\text{expl}}$
3: **while** training world model **do**
4:     **if** $p \sim U_{[0,1]} < p_{\text{DR}}$ or $D$.is_empty() **then**
5:         $\theta \sim$ DomainRandomisation($\Theta$)
6:     **else**
7:         $\theta \sim$ Boltzmann(Normalize($D_{\text{error}}$), $\eta$)
8:     $\tau_\theta \leftarrow$ rollout of $\pi^{\text{expl}}$ in $\mathcal{P}_\theta$     *Collect real traj. for $\theta$*
9:     Add $\tau_\theta$ to $D$
10:    Train $W$ on $D$
11:    $\pi^{\text{expl}}, D_{\text{error}} \leftarrow$ Imagine($D, D_{\text{error}}, W, \pi^{\text{expl}}$)
12: **function** Imagine($D, D_{\text{error}}, W, \pi^{\text{expl}}$)
13:     **for** $i = 1, \ldots, K$ **do**
14:         $\tau_\theta \leftarrow D$.sample()     *Sample real trajectory*
15:         $Z_{\tau_\theta} = \{z_t\}_{t=0}^{|\tau_\theta|} \leftarrow q(\tau_\theta)$     *Encode latent states*
16:         $\widehat{\tau}_\theta \leftarrow$ rollout $\pi^{\text{expl}}$ for $h$ steps from $z \in Z_{\tau_\theta}$ in $W$
17:         $\delta_\theta \leftarrow$ Error estimate for $\theta$ via Eq. 10 on $\widehat{\tau}_\theta$
18:         Update $D_{\text{error}}$ with $\delta_\theta$ for $\theta$
19:         Train $\pi^{\text{expl}}$ on $\widehat{\tau}_\theta$
20:     **return** $\pi^{\text{expl}}, D_{\text{error}}$

---

Imagine updates $\pi^{\text{expl}}$ and $D_{\text{error}}$ using imagined rollouts (Line 11). For each imagined rollout, we first sample real trajectory $\tau_\theta \in D$ (Line 14). We encode $\tau_\theta$ into latent states $Z_{\tau_\theta}$ (Line 15). We then generate an imagined trajectory, $\widehat{\tau}_\theta$, by rolling out $\pi^{\text{expl}}$ using $\widehat{T}$ starting from an initial latent state $z \in Z_{\tau_\theta}$ (Line 16). Thus, $\widehat{\tau}_\theta$ corresponds to an imagined trajectory in the environment with parameter setting $\theta$, as illustrated in Figure 1b. We wish to estimate the world model error for environment parameter setting $\theta$ from this imagined rollout. Following previous works (Mendonca et al., 2021; Rigter et al., 2023; Sekar et al., 2020), we learn an ensemble of $N$ latent dynamics models: $\{\widehat{T}_i\}_{i=1}^N$. Like (Yu et al., 2020; Kidambi et al., 2020; Lu et al., 2022), we use the disagreement between the ensemble means as an approximation to the world model TV error for parameters $\theta$:

$$\text{WORLDMODELERROR}(\widehat{T}, \theta) \approx \mathbb{E}_{z,a \sim d(\pi_\theta^{\text{expl}}, \widehat{\mathcal{M}}_\theta)}\left[\text{Var}\left\{\mathbb{E}[\widehat{T}_i(\cdot|z,a)]\right\}_{i=1}^N\right] \qquad (10)$$

We compute the error estimate using the latent states and actions in $\widehat{\tau}_\theta$ in Line 17. We use this to update our estimate of the world model error for environment $\theta$ in $D_{\text{error}}$ (Line 18). Optionally, we may also use the imagined rollout to update the exploration policy in Line 19. For the world model architecture, we use DreamerV2 (Hafner et al., 2021). Implementation details are in Appendix E.

**Error Estimate Update Function** We consider two possibilities for updating the error estimate for each $\theta$ in $D_{\text{error}}$ on Line 18: 1) in WAKER-M, $D_{\text{error}}$ maintains a smoothed average of the *magnitude* of the error estimate for each $\theta$; 2) In WAKER-R, we update $D_{\text{error}}$ to maintain a smoothed average

of the *rate of reduction* in the error estimate for each $\theta$. Thus, WAKER-M biases sampling towards environments that have highest error, while WAKER-R biases sampling towards environments that have the highest rate of reduction in error. More details are in Appendix E.

**Exploration Policy**   We must learn an exploration policy $\pi^{\text{expl}}$ that approximately maximises the world model error according to Equation 5. By default, we use Plan2Explore (Sekar et al., 2020), and train the exploration policy to maximise the approximation in Equation 10. To test whether our approach is agnostic to the exploration policy used, we also consider a random exploration policy.

**Zero-Shot Task Adaptation**   Once the world model has been trained, we use it to derive a task-specific policy without any further data collection. For reward function $R$, we find a single task policy $\widehat{\pi}^*_R$ that is defined over the entire latent space $Z$, and therefore all environments. We use the same approach as Sekar et al. (2020): we label the data in $D$ with the associated rewards and use this data to train a reward predictor. Then, we train the task policy in the world model to optimise the expected reward according to the reward predictor. All task policies are trained in this manner.

## 4   EXPERIMENTS

We seek to answer the following questions: a) Does WAKER enable more robust policies to be trained in the world model? b) Does the performance of WAKER depend on the exploration policy used? c) Does WAKER lead to stronger generalisation to out-of-distribution environments? The code for our experiments is available at github.com/marc-rigter/waker.

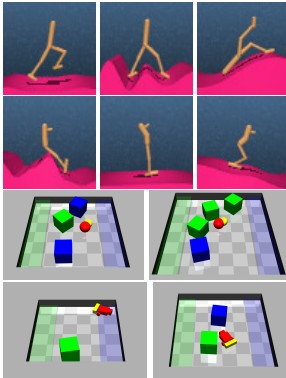

Figure 2: Example training environments. Rows: Terrain Walker, Terrain Hopper, Clean Up, Car Clean Up.

**Methods Compared**   To answer question a), we compare WAKER-M and WAKER-R with the following baselines: *Domain Randomisation* (DR), samples uniformly from the default environment distribution; *Gradual Expansion* (GE) gradually increases the range of environments sampled from the default distribution; *Hardest Environment Oracle* (HE-Oracle) samples only the single most complex environment; *Re-weighting Oracle* (RW-Oracle) re-weights the environment distribution to focus predominantly on the most complex environments. Note that the HE-Oracle and RW-Oracle baselines require expert domain knowledge. Detailed descriptions of the baselines can be found in Appendix G.

For both WAKER variants, we set $p_{\text{DR}} = 0.2$ and perform limited tuning of the Boltzmann temperature $\eta$. More details are in Appendix F.3. To investigate question b), we pair WAKER and DR with two different exploration policies: Plan2Explore (P2E) (Sekar et al., 2020) or a random exploration policy. For the other baselines we always use the P2E exploration policy.

**Domains and Tasks**   All domains use image observations. For *Terrain Walker* and *Terrain Hopper* we simulate the Walker and Hopper robots from the DMControl Suite (Tassa et al., 2018) on procedurally generated terrain. For each environment, there are two parameters (amplitude and length scale) that control the terrain generation. For each domain, we evaluate a number of downstream tasks (*walk*, *run*, *stand*, *flip*, *walk-back.* / *hop*, *hop-back.*, *stand*). The *Clean Up* and *Car Clean Up* domains are based on SafetyGym (Ray et al., 2019) and consist of blocks that can be pushed and either a point mass (*Clean Up*) or car robot (*Car Clean Up*). There are three environment parameters: the environment size, the number of blocks, and the block colours. The downstream tasks (*sort*, *push*, *sort-reverse*) each correspond to different goal locations for each colour of block. Examples of training environments are shown in Figure 2, and more details are in Appendix F.1. We also perform experiments where we train a *single world model* for both the Clean Up and Terrain Walker domains. Due to space constraints we defer these results to Appendix C.1.

**Evaluation**   Our aim is to train the world model such that the policies obtained are robust, as measured by minimax regret. However, we cannot directly evaluate regret as it requires knowing the true optimal performance. Therefore, following (Jiang et al., 2021a; Rigter et al., 2023) we evaluate conditional value at risk (CVaR) (Rockafellar et al., 2000) to measure robustness. For confidence level $\alpha$, $\text{CVaR}_\alpha$ is the average performance on the worst $\alpha$-fraction of runs. We evaluate $\text{CVaR}_{0.1}$ by evaluating each policy on 100 environments sampled uniformly at random, and reporting the aver-

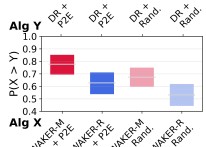

| Exploration Policy | | **Plan2Explore** | | | | | | **Random Exploration** | | |
|---|---|---|---|---|---|---|---|---|---|---|
| Environment Sampling | | WAKER-M | WAKER-R | DR | GE | HE-Oracle | RW-Oracle | WAKER-M | WAKER-R | DR |
| Clean Up | Sort | 0.711 ± 0.09 | 0.643 ± 0.07 | 0.397 ± 0.12 | 0.426 ± 0.09 | 0.240 ± 0.13 | 0.482 ± 0.14 | 0.007 ± 0.01 | 0.010 ± 0.01 | 0.000 ± 0.0 |
| | Sort-Rev. | 0.741 ± 0.06 | 0.586 ± 0.06 | 0.395 ± 0.10 | 0.490 ± 0.07 | 0.230 ± 0.11 | 0.537 ± 0.10 | 0.000 ± 0.0 | 0.000 ± 0.0 | 0.000 ± 0.0 |
| | Push | 0.716 ± 0.11 | 0.702 ± 0.08 | 0.590 ± 0.093 | 0.628 ± 0.12 | 0.262 ± 0.16 | 0.596 ± 0.14 | 0.124 ± 0.09 | 0.058 ± 0.06 | 0.023 ± 0.03 |
| Car Clean Up | Sort | 0.894 ± 0.04 | 0.815 ± 0.11 | 0.665 ± 0.14 | 0.641 ± 0.09 | 0.041 ± 0.07 | 0.624 ± 0.15 | 0.433 ± 0.11 | 0.378 ± 0.09 | 0.337 ± 0.07 |
| | Sort-Rev. | 0.914 ± 0.079 | 0.880 ± 0.080 | 0.659 ± 0.16 | 0.646 ± 0.13 | 0.043 ± 0.06 | 0.567 ± 0.16 | 0.408 ± 0.10 | 0.408 ± 0.09 | 0.269 ± 0.11 |
| | Push | 0.906 ± 0.05 | 0.888 ± 0.04 | 0.796 ± 0.10 | 0.807 ± 0.12 | 0.046 ± 0.06 | 0.777 ± 0.11 | 0.584 ± 0.12 | 0.526 ± 0.15 | 0.373 ± 0.12 |
| Terrain Walker | Walk | 818.0 ± 15.3 | 805.3 ± 42.0 | 748.9 ± 39.5 | 741.2 ± 43.6 | 543.2 ± 85.3 | 791.6 ± 32.3 | 243.9 ± 26.7 | 224.8 ± 41.9 | 224.3 ± 25.3 |
| | Run | 312.6 ± 19.9 | 303.0 ± 16.1 | 279.9 ± 18.1 | 300.1 ± 17.4 | 223.3 ± 18.6 | 305.5 ± 15.2 | 120.4 ± 14.7 | 104.2 ± 9.7 | 114.1 ± 12.2 |
| | Flip | 955.0 ± 11.9 | 937.7 ± 10.5 | 936.1 ± 10.2 | 946.0 ± 9.5 | 962.9 ± 5.7 | 952.4 ± 11.6 | 878.9 ± 18.4 | 850.7 ± 40.5 | 849.9 ± 27.4 |
| | Stand | 941.2 ± 12.3 | 945.4 ± 16.6 | 936.5 ± 17.5 | 938.6 ± 16.3 | 829.3 ± 66.5 | 923.4 ± 22.0 | 585.1 ± 31.8 | 581.5 ± 68.8 | 591.3 ± 65.5 |
| | Walk-Back. | 752.5 ± 24.8 | 722.1 ± 33.5 | 729.6 ± 39.2 | 700.4 ± 23.7 | 418.5 ± 94.3 | 712.2 ± 18.7 | 369.9 ± 13.1 | 311.5 ± 49.8 | 311.2 ± 48.4 |
| Terrain Hopper | Hop | 342.0 ± 35.2 | 301.3 ± 42.1 | 278.7 ± 43.0 | 267.6 ± 48.6 | 222.8 ± 23.1 | 345.5 ± 29.2 | 8.6 ± 7.4 | 9.1 ± 6.9 | 10.2 ± 7.4 |
| | Hop-Back. | 330.7 ± 24.9 | 284.3 ± 27.6 | 299.1 ± 26.6 | 285.6 ± 36.6 | 204.1 ± 27.3 | 324.0 ± 41.7 | 2.9 ± 5.6 | 2.7 ± 3.1 | 12.0 ± 13.3 |
| | Stand | 639.8 ± 68.3 | 699.0 ± 76.4 | 661.9 ± 51.5 | 625.0 ± 81.0 | 507.2 ± 89.7 | 656.6 ± 82.1 | 9.0 ± 7.2 | 25.7 ± 27.3 | 18.0 ± 10.2 |

Table 1: Robustness evaluation: $\text{CVaR}_{0.1}$ of policies evaluated on 100 randomly sampled environments.

Figure 3: Robustness evaluation aggregated CIs.

| Exploration Policy | **Plan2Explore** | | | | | | **Random Exploration** | | |
|---|---|---|---|---|---|---|---|---|---|
| Environment Sampling | WAKER-M | WAKER-R | DR | GE | HE-Oracle | RW-Oracle | WAKER-M | WAKER-R | DR |
| Clean Up Extra Block | 0.530 ± 0.04 | 0.479 ± 0.03 | 0.329 ± 0.04 | 0.348 ± 0.04 | 0.475 ± 0.03 | 0.515 ± 0.02 | 0.169 ± 0.02 | 0.132 ± 0.04 | 0.093 ± 0.02 |
| Car Clean Up Extra Block | 0.767 ± 0.03 | 0.713 ± 0.02 | 0.598 ± 0.02 | 0.582 ± 0.05 | 0.676 ± 0.04 | 0.695 ± 0.05 | 0.539 ± 0.02 | 0.510 ± 0.04 | 0.418 ± 0.04 |
| Terrain Walker Steep | 660.2 ± 13.6 | 647.7 ± 23.6 | 602.8 ± 12.8 | 605.2 ± 13.9 | 665.2 ± 16.1 | 673.6 ± 14.7 | 448.4 ± 25.4 | 387.4 ± 40.0 | 400.9 ± 17.7 |
| Terrain Walker Stairs | 665.7 ± 9.9 | 672.6 ± 17.4 | 622.9 ± 21.9 | 637.2 ± 18.2 | 641.6 ± 16.2 | 684.1 ± 23.9 | 451.1 ± 22.6 | 419.1 ± 29.3 | 414.5 ± 13.5 |
| Terrain Hopper Steep | 322.1 ± 18.4 | 299.0 ± 14.8 | 259.6 ± 27.3 | 232.9 ± 31.1 | 296.5 ± 22.6 | 299.9 ± 19.8 | 21.7 ± 8.4 | 19.4 ± 6.0 | 19.6 ± 5.3 |
| Terrain Hopper Stairs | 398.2 ± 16.4 | 416.6 ± 27.5 | 403.8 ± 11.4 | 385.7 ± 22.7 | 334.3 ± 15.8 | 395.8 ± 12.3 | 34.2 ± 13.5 | 31.4 ± 8.6 | 35.0 ± 10.7 |

Table 2: Out-of-distribution evaluation: average performance on OOD environments. Here, we present the average performance across tasks for each domain. Full results for each task are in Table 4 in Appendix C.2.

Figure 4: OOD evaluation aggregated CIs.

age of the worst 10 runs. We also report the average performance over all 100 runs in Appendix C.4. To answer question c), we additionally evaluate the policies on out-of-distribution (OOD) environments. For the terrain domains, the OOD environments are terrain with a length scale 25% shorter than seen in training (*Steep*), and terrain containing stairs (*Stairs*). For the clean up domains, the OOD environments contain one more block than was ever seen in training (*Extra Block*).

**Results Presentation** In Tables 1-2 we present results for task policies obtained from the *final world model* at the end of six days of training. For each exploration policy, we highlight results within 2% of the best score (provided that non-trivial performance is obtained), and ± indicates the S.D. over 5 seeds. *Learning curves* for the performance of policies obtained from snapshots of the world model are in Appendix D.2. To assess statistical significance, we present 95% confidence intervals of the probability that algorithm X obtains improved performance over algorithm Y, computed using the aggregated results across all tasks with the *rliable* (Agarwal et al., 2021) framework (Figures 3-4).

**Results** Table 1 presents the robustness evaluation results. For both exploration policies, WAKER-M outperforms DR across almost all tasks. For the Plan2Explore exploration policy, WAKER-R also outperforms DR. Figure 3 shows that these improvements over DR are statistically significant, as the lower bound on the probability of improvement is greater than 0.5. Both WAKER variants result in significant improvements over DR when using Plan2Explore, suggesting that WAKER is more effective when combined with a sophisticated exploration policy. Between WAKER-M and WAKER-R, WAKER-M (which prioritises the most uncertain environments) obtains stronger performance. This is expected from our analysis in Section 3.3, which shows that minimising the maximum world model error across environments improves robustness. Figure 5 illustrates that WAKER focuses sampling on more complex environments: larger environments with more blocks in the clean up domains, and steeper terrain with shorter length scale and greater amplitude in the terrain domains. Plots of how the sampling evolves during training are in Appendix C.3. Regardless of the environment selection method, Plan2Explore leads to stronger performance than random exploration, verifying previous findings (Sekar et al., 2020). The results for average performance in Appendix C.4 show that WAKER achieves improved or similar average performance compared to DR. This demonstrates that WAKER *improves robustness at no cost* to average performance. The results for training a *single world model* for both the Clean Up and Terrain Walker environments in Appendix C.1 demonstrate that even when training across highly diverse environments, WAKER also achieves more robust performance in comparison to DR.

The robustness results in Table 1 show that GE obtains very similar performance to DR. This is unsurprising, as GE does not bias sampling towards more difficult environments, and only modifies the DR distribution by gradually increasing the range of environments sampled. HE-Oracle obtains poor performance, demonstrating that focussing on the most challenging environment alone is insufficient to obtain a robust world model. This is expected from the analysis in Section 3.3 which shows that to obtain robustness we need the world model to have low error across all environments (not just the most complex one). For Terrain Walker and Terrain Hopper, RW-Oracle is a strong baseline, and obtains similar performance to WAKER. However, for Clean Up and Car Clean Up RW-Oracle

obtains significantly worse performance than WAKER. This is likely because by focussing sampling environments with four blocks of any colour, RW-Oracle does not sample diversely enough to obtain good robustness. This demonstrates that *even with domain knowledge, handcrafting a suitable curriculum is challenging.*

To assess the quality of the world models, we evaluate the error between the transitions predicted by the world model and real transitions. To evaluate robustness, we compute the error on the worst 10% of trajectories generated by a performant task-policy on randomly sampled environments. In Figure 6, we observe that WAKER leads to lower prediction errors on the worst 10% of trajectories compared to DR. This verifies that by biasing sampling towards environments with higher error estimates, WAKER reduces the worst-case errors in the world model. This suggests that improved world model training leads to the improved policy robustness that we observe in Table 1, as expected from the upper bound in Equation 7.

Table 2 and Figure 4 present the evaluation on OOD environments, averaged across tasks. Full results for each task are in Appendix C.2. WAKER-M results in a considerable performance improvement over DR for both exploration policies, and WAKER-R significantly improves performance when the exploration policy is Plan2Explore. Thus, WAKER-M again obtains better performance than WAKER-R, but both variants of our algorithm outperform DR. GE obtains similar or slightly worse performance than DR for the OOD environments. For the steep terrain OOD environments, HE-Oracle performs quite well as it focuses on sampling the steepest possible in-distribution terrain. However, HE-Oracle does not perform well on the stairs OOD environments, demonstrating that sampling a range of environments is necessary for strong OOD generalisation. RW-Oracle performs well on the OOD environments across all domains. However, RW-Oracle has the significant drawback that expert domain knowledge is required. These results demonstrate that by actively sampling more uncertain environments for exploration, WAKER leads to world models that are able to generalise more broadly to environments never seen during training, without requiring any expert domain knowledge.

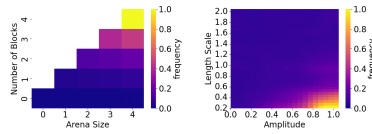

(a) Clean Up  (b) Terrain Walker

Figure 5: Heatmaps of environment parameters sampled by WAKER-M + P2E.

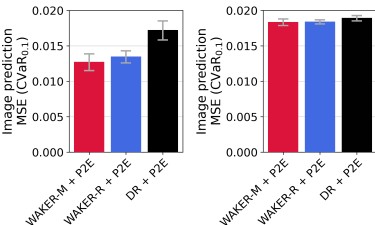

(a) Car Clean Up  (b) Terrain Hopper

Figure 6: Evaluation of $CVaR_{0.1}$ of the world model image prediction error.

## 5 CONCLUSION

In this work, we have proposed the first method for automatic curriculum learning for environment selection in the reward-free setting. Due to space constraints, a discussion of related work can be found in Appendix A. We formalised this problem in the context of learning a robust world model. Building on prior works in robust RL, we considered robustness in terms of minimax regret, and we derived a connection between the maximum regret and the maximum error of the world model dynamics across environments. We operationalised this insight in the form of WAKER, which trains a world model in the reward-free setting by selectively sampling the environment settings that induce the highest latent dynamics error. In several pixel-based continuous control domains, we demonstrated that compared to other baselines that do not require expert domain knowledge, WAKER drastically improves the zero-shot task adaptation capabilities of the world model in terms of robustness. Policies trained for downstream tasks inside the learned world model exhibit significantly improved generalisation to out-of-distribution environments that were never encountered during training. WAKER therefore represents a meaningful step towards developing more generally-capable agents. In future work, we are excited to scale our approach to even more complex domains with many variable parameters. One limitation of our approach is that it relies upon an intrinsically motivated policy to adequately explore the state-action space across a range of environments. This may pose a challenge for scalability to more complex environments. To scale WAKER further, we plan to make use of function approximation to estimate uncertainty throughout large parameter spaces, as opposed to the discrete buffer used in this work. We also plan to investigate using WAKER for reward-free pretraining, followed by task-specific finetuning to overcome the challenge of relying upon intrinsically motivated exploration.

**Acknowledgements** This work was supported by a Programme Grant from the Engineering and Physical Sciences Research Council (EP/V000748/1) and a gift from Amazon Web Services. Additionally, this project made use of the Tier 2 HPC facility JADE2, funded by the Engineering and Physical Sciences Research Council (EP/T022205/1).

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

## A    RELATED WORK

Our work focuses on training more robust world-models (Ha & Schmidhuber, 2018; Hafner et al., 2020; 2019; 2021; Janner et al., 2019; Rigter et al., 2022; Schrittwieser et al., 2020) in the reward-free setting. Past works improve data collection by training an exploration policy within the world model (i.e. in imagination) (Sekar et al., 2020; Ball et al., 2020; Mendonca et al., 2021; Xu et al., 2022). Unlike prior methods, WAKER augments action-based exploration by directly exploring over the space of environments via an autocurriculum. To our knowledge, this is the first work to address curricula for reward-free training of a world model.

WAKER can thus be viewed as an *automatic curriculum learning* (ACL) (Graves et al., 2017; Portelas et al., 2021; Narvekar et al., 2020) method for training a world model. ACL methods optimise the order in which tasks or environments are presented during training, to improve the agent's final performance (Asada et al., 1996; Florensa et al., 2017; 2018; Mendonca et al., 2021). Existing ACL approaches for environment selection require *task-specific* evaluation metrics to assess the performance or learning potential of the agent in each environment, and select appropriate environments to sample from (Portelas et al., 2020; Eimer et al., 2021; Dennis et al., 2020; Jiang et al., 2021a; Matiisen et al., 2019; Mehta et al., 2020; Parker-Holder et al., 2022; Wang et al., 2019). Therefore, existing ACL approaches are not applicable to the reward-free setting that we address.

Domain randomisation (DR) (Jakobi, 1997; Tobin et al., 2017) samples environments uniformly, and can be viewed as the simplest approach to ACL. Active DR (Mehta et al., 2020; Akkaya et al., 2019) extends DR by upweighting environments that induce divergent behaviors relative to a reference behaviour. However, Active DR requires access to a task-specific reward function to generate the reference behaviour and therefore is also not applicable to our reward-free setting.

*Unsupervised environment design* (UED) (Dennis et al., 2020) refers to a class of methods that generate curricula to produce more robust policies. UED typically frames curriculum learning as a zero-sum game between a teacher and a student, where the teacher actively selects environments to optimise an adversarial objective. When this objective is the student's regret in each environment, the student provably follows a minimax regret policy at the Nash equilibria of the resulting game, making minimax regret UED methods a principled approach for learning robust policies (Dennis et al., 2020; Jiang et al., 2021a). A simple and effective form of UED is Prioritised Level Replay (PLR) (Jiang et al., 2021b;a), which selectively *revisits* environments that have higher estimated "learning potential", as measured by the temporal difference (TD) error. Extensions of PLR improve on its random search (Parker-Holder et al., 2022) and address issues with UED in stochastic settings (Jiang et al., 2022a). However, all existing UED methods require a known reward function during curriculum learning. To our knowledge, ours is the first work to address reward-free UED.

WAKER uses ideas from *intrinsic motivation* (Deci & Ryan, 1985; Schmidhuber, 2010) as a bridge between UED and the reward-free setting. By providing agents with *intrinsic rewards*, such methods enable agents to efficiently explore complex environments in the absence of extrinsic rewards. Intrinsic motivation generalises active learning (Cohn et al., 1996) to sequential decision-making, by rewarding agents for visiting states that: a) have high uncertainty (Haber et al., 2018; Pathak et al., 2017; Schmidhuber, 1991; Sekar et al., 2020; Shyam et al., 2019), b) can be maximally influenced (Bharadhwaj et al., 2022; Klyubin et al., 2005), or c) have rarely been visited (Bellemare et al., 2016; Ostrovski et al., 2017). Our work extends uncertainty-based intrinsic motivation beyond states within a single environment to the space of environment configurations, allowing novelty-related metrics to be used in place of typical UED objectives that require a task-specific reward function. WAKER thus highlights the fundamental connection between autocurricula and exploration.

## B    PROOF OF PROPOSITION 1

We begin by stating and proving a version of the classic Simulation Lemma from (Kearns & Singh, 2002).

**Lemma 1 (Simulation Lemma (Kearns & Singh, 2002))** *Consider two infinite horizon MDPs,* $\mathcal{M}^R = \{S, A, R, T, \gamma\}$ *and* $\widehat{\mathcal{M}}^R = \{S, A, R, \widehat{T}, \gamma\}$*, with reward function* $R : S \times A \rightarrow [0, 1]$.

*Consider any stochastic policy $\pi : S \rightarrow \Delta(A)$. Then:*

$$\left| V(\pi, \widehat{\mathcal{M}}^R) - V(\pi, \mathcal{M}^R) \right| \leq \frac{2\gamma}{(1-\gamma)^2} \mathbb{E}_{s,a \sim d(\pi, \widehat{\mathcal{M}}^R)} \left[ \text{TV} \left( \widehat{T}(\cdot|s,a), T(\cdot|s,a) \right) \right] \qquad (11)$$

*where $d(\pi, \widehat{\mathcal{M}}^R)$ is the state-action distribution of $\pi$ in $\widehat{\mathcal{M}}^R$, and $\text{TV}(P, Q)$ is the total variation distance between two distributions $P$ and $Q$.*

*Proof*: To simplify the notation of the proof, we will use the notation $V^\pi = V(\pi, \mathcal{M}^R)$ and $\widehat{V}^\pi = V(\pi, \widehat{\mathcal{M}}^R)$.

Using the Bellman equation for $\widehat{V}^\pi$ and $V^\pi$, we have:

$$\widehat{V}^\pi(s_0) - V^\pi(s_0) = \mathbb{E}_{a_0 \sim \pi(s_0)} \left[ R(s_0, a_0) + \gamma \mathbb{E}_{s' \sim \widehat{T}(s_0, a_0)} \widehat{V}^\pi(s') \right]$$

$$- \mathbb{E}_{a_0 \sim \pi(s_0)} \left[ R(s_0, a_0) + \gamma \mathbb{E}_{s' \sim T(s_0, a_0)} V^\pi(s') \right]$$

$$= \gamma \mathbb{E}_{a_0 \sim \pi(s_0)} \left[ \mathbb{E}_{s' \sim \widehat{T}(s_0, a_0)} \widehat{V}^\pi(s') - \mathbb{E}_{s' \sim T(s_0, a_0)} V^\pi(s') \right]$$

$$= \gamma \mathbb{E}_{a_0 \sim \pi(s_0)} \left[ \mathbb{E}_{s' \sim \widehat{T}(s_0, a_0)} \widehat{V}^\pi(s') - \mathbb{E}_{s' \sim \widehat{T}(s_0, a_0)} V^\pi(s') \right.$$

$$\left. + \mathbb{E}_{s' \sim \widehat{T}(s_0, a_0)} V^\pi(s') - \mathbb{E}_{s' \sim T(s_0, a_0)} V^\pi(s') \right]$$

$$= \gamma \mathbb{E}_{a_0 \sim \pi(s_0)} \left[ \mathbb{E}_{s' \sim \widehat{T}(s_0, a_0)} \widehat{V}^\pi(s') - \mathbb{E}_{s' \sim \widehat{T}(s_0, a_0)} V^\pi(s') \right]$$

$$+ \gamma \mathbb{E}_{a_0 \sim \pi(s_0)} \left[ \mathbb{E}_{s' \sim \widehat{T}(s_0, a_0)} V^\pi(s') - \mathbb{E}_{s' \sim T(s_0, a_0)} V^\pi(s') \right]$$

$$(12)$$

We define $\widehat{P}_1^\pi(s_1|s_0) = \int_{a \in A} \pi(a|s_0) \widehat{T}(s_1|s_0, a) \mathrm{d}a$. This allows us to rewrite the first term from the last line as:

$$\gamma \mathbb{E}_{a_0 \sim \pi(s_0)} \left[ \mathbb{E}_{s' \sim \widehat{T}(s_0, a_0)} \widehat{V}^\pi(s') - \mathbb{E}_{s' \sim \widehat{T}(s_0, a_0)} V^\pi(s') \right] = \gamma \mathbb{E}_{s_1 \sim \widehat{P}_1^\pi(\cdot|s_0)} \left[ \widehat{V}^\pi(s_1) - V^\pi(s_1) \right]$$

We define $\widehat{P}_1^\pi(s_1, a_1|s_0) = \pi(a_1|s_1) \widehat{P}_1^\pi(s_1|s_0)$, and $\widehat{P}_2^\pi(s_2|s_0) = \int_{s \in S} \int_{a \in A} \widehat{P}_1^\pi(s, a|s_0) \widehat{T}(s_2|s, a) \mathrm{d}a \mathrm{d}s$. We again apply the Bellman equation:

$$\gamma \mathbb{E}_{s_1 \sim \widehat{P}_1^\pi(\cdot|s_0)} \left[ \widehat{V}^\pi(s_1) - V^\pi(s_1) \right] =$$

$$\gamma \mathbb{E}_{s_1 \sim \widehat{P}_1^\pi(\cdot|s_0)} \left[ \gamma \mathbb{E}_{a_1 \sim \pi(\cdot|s_1)} \left[ \mathbb{E}_{s' \sim \widehat{T}(s_1, a_1)} \widehat{V}^\pi(s') - \mathbb{E}_{s' \sim T(s_1, a_1)} V^\pi(s') \right] \right]$$

$$= \gamma \mathbb{E}_{s_1 \sim \widehat{P}_1^\pi(\cdot|s_0)} \left[ \gamma \mathbb{E}_{a_1 \sim \pi(\cdot|s_1)} \left[ \mathbb{E}_{s' \sim \widehat{T}(s_1, a_1)} \widehat{V}^\pi(s') - \mathbb{E}_{s' \sim \widehat{T}(s_1, a_1)} V^\pi(s') \right] \right] +$$

$$\gamma \mathbb{E}_{s_1 \sim \widehat{P}_1^\pi(\cdot|s_0)} \left[ \gamma \mathbb{E}_{a_1 \sim \pi(\cdot|s_1)} \left[ \mathbb{E}_{s' \sim \widehat{T}(s_1, a_1)} V^\pi(s') - \mathbb{E}_{s' \sim T(s_1, a_1)} V^\pi(s') \right] \right]$$

$$= \gamma^2 \mathbb{E}_{s_2 \sim \widehat{P}_2^\pi(\cdot|s_0)} \left[ \widehat{V}^\pi(s_2) - V^\pi(s_2) \right] + \gamma \mathbb{E}_{s_1, a_1 \sim \widehat{P}_1^\pi(\cdot, \cdot|s_0)} \left[ \mathbb{E}_{s' \sim \widehat{T}(s_1, a_1)} V^\pi(s') - \mathbb{E}_{s' \sim T(s_1, a_1)} V^\pi(s') \right].$$

$$(13)$$

Combining Equations 12 and 13 we have

$$\widehat{V}^\pi(s_0) - V^\pi(s_0) = \gamma \mathbb{E}_{a_0 \sim \pi(s_0)} \left[ \mathbb{E}_{s' \sim \widehat{T}(s_0, a_0)} V^\pi(s') - \mathbb{E}_{s' \sim T(s_0, a_0)} V^\pi(s') \right] +$$

$$\gamma \mathbb{E}_{s_1, a_1 \sim \widehat{P}_1^\pi(\cdot, \cdot|s_0)} \left[ \mathbb{E}_{s' \sim \widehat{T}(s_1, a_1)} V^\pi(s') - \mathbb{E}_{s' \sim T(s_1, a_1)} V^\pi(s') \right] +$$

$$\gamma^2 \mathbb{E}_{s_2 \sim \widehat{P}_2^\pi(\cdot|s_0)} \left[ \widehat{V}^\pi(s_2) - V^\pi(s_2) \right] \quad (14)$$

Repeatedly applying the reasoning in Equations 12-14 we have

$$
\begin{aligned}
\widehat{V}^\pi(s_0) - V^\pi(s_0) &= \sum_{t=0}^{\infty} \gamma^{t+1} \mathbb{E}_{s,a \sim \widehat{P}_t^\pi(\cdot,\cdot|s_0)} \left[ \mathbb{E}_{s' \sim \widehat{T}(s,a)} V^\pi(s_t) - \mathbb{E}_{s' \sim T(s,a)} V^\pi(s_t) \right] \\
&= \frac{\gamma}{1-\gamma} \mathbb{E}_{s,a \sim d(\pi, \widehat{\mathcal{M}}^R)} \left[ \mathbb{E}_{s' \sim \widehat{T}(s,a)} V^\pi(s_t) - \mathbb{E}_{s' \sim T(s,a)} V^\pi(s_t) \right]
\end{aligned}
\tag{15}
$$

where $d(\pi, \widehat{\mathcal{M}}^R)$ is the state-action distribution of $\pi$ in $\widehat{\mathcal{M}}^R$. By the definition of the reward function ($R(s,a) \in [0,1]$), we have that $V^\pi(s) \in [0, 1/(1-\gamma)]$. We utilise the inequality that: $|\mathbb{E}_{x \sim P} f(x) - \mathbb{E}_{x \sim Q} f(x)| \leq 2 \max_x |f(x)| \mathrm{TV}(P, Q)$. Then, taking the absolute value of Equation 15 and applying the inequality we have:

$$
\begin{aligned}
\left| \widehat{V}^\pi(s_0) - V^\pi(s_0) \right| &\leq \frac{\gamma}{1-\gamma} \mathbb{E}_{s,a \sim d(\pi, \widehat{\mathcal{M}}^R)} \left| \mathbb{E}_{s' \sim \widehat{T}(s,a)} V^\pi(s_t) - \mathbb{E}_{s' \sim T(s,a)} V^\pi(s_t) \right| \\
&\leq \frac{2\gamma}{(1-\gamma)^2} \mathbb{E}_{s,a \sim d(\pi, \widehat{\mathcal{M}}^R)} \left[ \mathrm{TV}\big(\widehat{T}(\cdot|s,a), T(\cdot|s,a)\big) \right] \quad \square
\end{aligned}
\tag{16}
$$

Now that we have proven Lemma 1, we return to our original purpose of proving Proposition 1. We begin by restating Proposition 1:

**Proposition 1** *Let $\widehat{T}$ be the learnt latent dynamics in the world model. Assume the existence of a representation model q that adheres to Assumption 2, and let T be the true latent dynamics according to Assumption 2. Then, for any parameter setting $\theta$ and reward function R, the regret of the optimal world model policy is bounded according to:*

$$
\begin{aligned}
\mathrm{REGRET}(\widehat{\pi}_{\theta,R}^*, \mathcal{P}_\theta^R) \leq \frac{2\gamma}{(1-\gamma)^2} \Big[ &\mathbb{E}_{z,a \sim d(\pi_{\theta,R}^*, \widehat{\mathcal{M}}_\theta)} \big[ \mathrm{TV}\big(\widehat{T}(\cdot|z,a), T(\cdot|z,a)\big) \big] \\
&+ \mathbb{E}_{z,a \sim d(\widehat{\pi}_{\theta,R}^*, \widehat{\mathcal{M}}_\theta)} \big[ \mathrm{TV}\big(\widehat{T}(\cdot|z,a), T(\cdot|z,a)\big) \big] \Big]
\end{aligned}
$$

*where $d(\pi, \mathcal{M})$ denotes the state-action distribution of $\pi$ in MDP $\mathcal{M}$, and $\mathrm{TV}(P, Q)$ is the total variation distance between distributions P and Q.*

Now, let us consider the latent MDP learnt by our world model $\widehat{\mathcal{M}}_\theta = (Z_\theta, A, \widehat{T}, \gamma)$ for some parameters $\theta \in \Theta$, as well as the latent space MDP that exactly models the true environment dynamics according to Assumption 2, $\mathcal{M}_\theta = (Z_\theta, A, T, \gamma)$. Recall that for some reward function, $R$, the optimal world model policy is defined to be

$$
\widehat{\pi}_{\theta,R}^* = \arg\max_\pi V(\pi, \widehat{\mathcal{M}}_\theta^R)
$$

The regret of the optimal world model policy for parameter setting $\theta$ and reward function $R$ is

$$
\mathrm{REGRET}(\widehat{\pi}_{\theta,R}^*, \mathcal{P}_\theta^R) = V(\pi_{\theta,R}^*, \mathcal{P}_\theta^R) - V(\widehat{\pi}_{\theta,R}^*, \mathcal{P}_\theta^R)
\tag{17}
$$

By Assumption 2 we have that $V(\pi, \mathcal{P}_\theta^R) = V(\pi, \mathcal{M}_\theta^R)$ for all $\pi$. This allows us to write the regret as the performance difference in $\mathcal{M}_\theta^R$ rather than $\mathcal{P}_\theta^R$:

$$
\mathrm{REGRET}(\widehat{\pi}_{\theta,R}^*, \mathcal{P}_\theta^R) = V(\pi_{\theta,R}^*, \mathcal{M}_\theta^R) - V(\widehat{\pi}_{\theta,R}^*, \mathcal{M}_\theta^R)
\tag{18}
$$

By the definition of the optimal world model policy, we have that

$$
V(\widehat{\pi}_{\theta,R}^*, \widehat{\mathcal{M}}_\theta^R) - V(\pi_{\theta,R}^*, \widehat{\mathcal{M}}_\theta^R) \geq 0
\tag{19}
$$

Adding together Equations 18 and 19 we have

$$
\begin{aligned}
\mathrm{REGRET}(\widehat{\pi}_{\theta,R}^*, \mathcal{P}_\theta^R) &\leq V(\pi_{\theta,R}^*, \mathcal{M}_\theta^R) - V(\pi_{\theta,R}^*, \widehat{\mathcal{M}}_\theta^R) + V(\widehat{\pi}_{\theta,R}^*, \widehat{\mathcal{M}}_\theta^R) - V(\widehat{\pi}_{\theta,R}^*, \mathcal{M}_\theta^R) \\
&\leq \left| V(\pi_{\theta,R}^*, \mathcal{M}_\theta^R) - V(\pi_{\theta,R}^*, \widehat{\mathcal{M}}_\theta^R) \right| + \left| V(\widehat{\pi}_{\theta,R}^*, \widehat{\mathcal{M}}_\theta^R) - V(\widehat{\pi}_{\theta,R}^*, \mathcal{M}_\theta^R) \right|
\end{aligned}
$$

Then, applying the Lemma 1 to both terms on the right-hand side we have

$$
\begin{aligned}
\mathrm{REGRET}(\widehat{\pi}_{\theta,R}^*, \mathcal{P}_\theta^R) \leq \frac{2\gamma}{(1-\gamma)^2} \Big[ &\mathbb{E}_{z,a \sim d(\pi_{\theta,R}^*, \widehat{\mathcal{M}}_\theta)} \big[ \mathrm{TV}\big(\widehat{T}(\cdot|z,a), T(\cdot|z,a)\big) \big] \\
&+ \mathbb{E}_{z,a \sim d(\widehat{\pi}_{\theta,R}^*, \widehat{\mathcal{M}}_\theta)} \big[ \mathrm{TV}\big(\widehat{T}(\cdot|z,a), T(\cdot|z,a)\big) \big] \Big] \square
\end{aligned}
$$

## C    Key Additional Results

In this section, we present the most important additional results that were omitted from the main paper due to space constraints.

### C.1    Training a Single World Model for Two Domains

In this subsection, we present results for training a *single world model* on both the Terrain Walker and Clean Up domains. WAKER must choose which domain to sample from (either Terrain Walker or Clean Up) as well as sample the environment parameters for that domain. The domain randomisation (DR) baseline chooses either domain with 50% probability, and then randomly samples the parameters for that domain.

To handle the varying dimensionality of the action spaces between the two domains, we pad the action space of the Clean Up domain with additional actions that are unused in that domain. To ensure that WAKER receives well-calibrated ensemble-based uncertainty estimates between these two significantly different domains, we use the following approach. We first normalise the uncertainty estimates for Terrain Walker and Clean Up separately, and then concatenate the uncertainty estimates to create the buffer of uncertainty estimates. We found that this was necessary because the scale of the ensemble-based uncertainty estimates can differ between the two significantly different domains. Obtaining well-calibrated uncertainty estimates without requiring this normalisation process is an orthogonal area of research that is outside the scope of this paper.

The results in Table 3 show that WAKER outperforms DR on both the robustness evaluation and the out-of-distribution evaluation when training a single world model on both domains. This highlights the potential for WAKER to be used to train very general world models, to enable agents capable of solving a wide range of tasks in a wide range of domains.

Table 3: Results for training a single world model for two domains (Clean Up + Terrain Walker) over five seeds. Evaluated after 8M total steps of reward-free training. For each episode, WAKER chooses the domain as well as the environment parameters for that domain. Domain randomisation (DR) randomly samples the domain and the parameters. Due to resource constraints, we evaluate a subset of the tasks.

(a) Robustness Evaluation ($\text{CVaR}_{0.1}$).

| Exploration Policy: | | **Plan2Explore** | |
| Env. Sampling: | | **WAKER-M** | **DR** |
|---|---|---|---|
| Clean Up | Sort | $0.348 \pm 0.080$ | $0.089 \pm 0.07$ |
| | Push | $0.436 \pm 0.141$ | $0.262 \pm 0.08$ |
| Terrain | Walk | $538.5 \pm 42.1$ | $499.1 \pm 43.6$ |
| Walker | Run | $195.4 \pm 8.2$ | $192.9 \pm 16.6$ |

(b) Out-of-distribution average performance.

| Exploration Policy: | | **Plan2Explore** | |
| Env. Sampling: | | **WAKER-M** | **DR** |
|---|---|---|---|
| Clean Up: | Sort | $0.333 \pm 0.03$ | $0.180 \pm 0.05$ |
| Extra Block | Push | $0.440 \pm 0.03$ | $0.294 \pm 0.04$ |
| Terrain Walker: | Walk | $420.5 \pm 33.4$ | $382.6 \pm 34.3$ |
| Steep | Run | $154.9 \pm 9.8$ | $149.6 \pm 13.9$ |
| Terrain Walker: | Walk | $417.1 \pm 55.1$ | $412.8 \pm 33.9$ |
| Stairs | Run | $161.8 \pm 17.6$ | $160.9 \pm 10.2$ |

### C.2    Out of Distribution Evaluation: Full Task Results

In Table 2 in the main paper, we presented the out-of-distribution results in terms of averages across the tasks for each domain. In Table 4 we present the full results for each task.

| Exploration Policy | | Plan2Explore | | | | | | Random Exploration | | |
|---|---|---|---|---|---|---|---|---|---|---|
| Environment | Sampling | WAKER-M | WAKER-R | DR | GE | HE-Oracle | RW-Oracle | WAKER-M | WAKER-R | DR |
| Clean Up Extra Block | Sort | 0.499 ± 0.04 | 0.486 ± 0.02 | 0.289 ± 0.05 | 0.310 ± 0.03 | 0.437 ± 0.03 | 0.501 ± 0.03 | 0.143 ± 0.03 | 0.107 ± 0.03 | 0.071 ± 0.03 |
| | Sort-Rev. | 0.524 ± 0.05 | 0.437 ± 0.05 | 0.295 ± 0.06 | 0.314 ± 0.04 | 0.458 ± 0.03 | 0.506 ± 0.03 | 0.140 ± 0.04 | 0.110 ± 0.04 | 0.066 ± 0.05 |
| | Push | 0.567 ± 0.02 | 0.515 ± 0.01 | 0.402 ± 0.05 | 0.419 ± 0.04 | 0.529 ± 0.04 | 0.537 ± 0.02 | 0.220 ± 0.05 | 0.180 ± 0.03 | 0.141 ± 0.04 |
| Car Clean Up Extra Block | Sort | 0.749 ± 0.05 | 0.660 ± 0.04 | 0.549 ± 0.04 | 0.545 ± 0.05 | 0.638 ± 0.05 | 0.671 ± 0.06 | 0.480 ± 0.05 | 0.482 ± 0.04 | 0.415 ± 0.06 |
| | Sort-Rev. | 0.775 ± 0.04 | 0.724 ± 0.06 | 0.572 ± 0.04 | 0.542 ± 0.04 | 0.647 ± 0.03 | 0.674 ± 0.04 | 0.503 ± 0.05 | 0.465 ± 0.05 | 0.374 ± 0.05 |
| | Push | 0.778 ± 0.04 | 0.754 ± 0.05 | 0.674 ± 0.05 | 0.659 ± 0.04 | 0.742 ± 0.05 | 0.740 ± 0.04 | 0.633 ± 0.03 | 0.582 ± 0.07 | 0.466 ± 0.12 |
| Terrain Walker Steep | Walk | 631.4 ± 20.8 | 583.8 ± 51.9 | 512.7 ± 31.2 | 539.8 ± 29.5 | 633.0 ± 14.9 | 644.4 ± 21.8 | 212.3 ± 20.2 | 190.3 ± 36.1 | 185.1 ± 26.5 |
| | Run | 228.6 ± 12.8 | 213.2 ± 17.2 | 186.8 ± 9.1 | 197.8 ± 7.6 | 222.6 ± 10.5 | 232.3 ± 5.6 | 119.3 ± 9.9 | 103.4 ± 10.0 | 105.2 ± 15.4 |
| | Flip | 946.3 ± 15.1 | 950.3 ± 8.0 | 929.1 ± 23.8 | 902.0 ± 18.4 | 978.5 ± 2.4 | 953.2 ± 15.8 | 891.5 ± 14.8 | 817.0 ± 59.0 | 885.4 ± 29.6 |
| | Stand | 940.8 ± 15.2 | 934.9 ± 21.2 | 919.4 ± 23.1 | 915.7 ± 21.2 | 956.3 ± 8.4 | 951.5 ± 8.5 | 690.6 ± 29.6 | 612.5 ± 57.5 | 625.5 ± 81.4 |
| | Walk-Back. | 554.0 ± 21.5 | 547.5 ± 39.4 | 465.9 ± 18.9 | 470.8 ± 33.4 | 535.5 ± 28.7 | 586.7 ± 21.9 | 328.3 ± 18.0 | 213.7 ± 72.7 | 203.3 ± 33.5 |
| Terrain Walker Stairs | Walk | 686.6 ± 38.4 | 668.4 ± 45.7 | 611.7 ± 75.2 | 625.7 ± 30.8 | 622.6 ± 57.2 | 691.2 ± 27.2 | 224.2 ± 27.3 | 217.3 ± 23.7 | 219.9 ± 17.0 |
| | Run | 241.2 ± 16.4 | 235.6 ± 14.1 | 216.1 ± 24.3 | 225.9 ± 9.3 | 225.0 ± 9.4 | 244.6 ± 8.5 | 128.2 ± 6.0 | 114.5 ± 11.0 | 119.5 ± 9.7 |
| | Flip | 935.9 ± 15.9 | 939.3 ± 8.8 | 927.6 ± 8.4 | 929.8 ± 7.1 | 957.1 ± 8.4 | 953.6 ± 6.7 | 892.0 ± 15.1 | 840.6 ± 39.7 | 822.5 ± 101.6 |
| | Stand | 972.8 ± 6.8 | 971.8 ± 7.6 | 970.2 ± 7.6 | 971.2 ± 6.9 | 960.7 ± 7.0 | 969.2 ± 10.2 | 697.7 ± 41.8 | 665.8 ± 72.2 | 668.1 ± 72.1 |
| | Walk-Back. | 492.1 ± 55.1 | 548.1 ± 51.3 | 388.7 ± 34.6 | 433.2 ± 16.6 | 442.6 ± 32.6 | 561.9 ± 19.6 | 313.6 ± 5.4 | 257.1 ± 21.3 | 242.6 ± 24.8 |
| Terrain Hopper Steep | Hop | 137.2 ± 21.5 | 112.9 ± 23.5 | 93.7 ± 14.6 | 85.7 ± 8.4 | 143.0 ± 16.6 | 116.9 ± 17.9 | 5.1 ± 2.6 | 9.4 ± 3.9 | 13.3 ± 5.7 |
| | Hop-Back. | 133.6 ± 13.5 | 104.7 ± 13.1 | 92.5 ± 13.9 | 77.5 ± 12.3 | 138.3 ± 12.6 | 110.7 ± 25.1 | 30.1 ± 14.9 | 19.7 ± 12.3 | 21.6 ± 10.5 |
| | Stand | 695.4 ± 29.4 | 679.7 ± 36.5 | 592.6 ± 74.6 | 535.5 ± 102.6 | 608.2 ± 64.0 | 672.0 ± 57.1 | 29.8 ± 10.2 | 29.1 ± 20.6 | 24.0 ± 12.6 |
| Terrain Hopper Stairs | Hop | 292.1 ± 16.6 | 258.1 ± 29.3 | 250.7 ± 24.9 | 220.0 ± 33.6 | 229.5 ± 14.4 | 263.3 ± 23.7 | 17.4 ± 13.3 | 25.6 ± 10.2 | 31.7 ± 14.7 |
| | Hop-Back. | 181.6 ± 21.8 | 209.1 ± 23.5 | 188.5 ± 25.7 | 164.3 ± 20.4 | 117.5 ± 18.8 | 177.5 ± 47.2 | 33.6 ± 9.6 | 19.5 ± 12.2 | 34.3 ± 18.5 |
| | Stand | 720.8 ± 47.6 | 782.5 ± 59.7 | 772.2 ± 42.9 | 772.8 ± 30.8 | 655.8 ± 54.3 | 746.7 ± 47.5 | 51.5 ± 32.6 | 49.1 ± 29.4 | 39.2 ± 17.1 |

Table 4: Out-of-distribution evaluation: average performance on OOD environments for each domain and task. For Terrain Hopper with a random exploration policy, all sampling methods fail to learn a meaningful policy so none are highlighted.

## C.3 ILLUSTRATION OF CURRICULA

The plots in Figures 7 and 8 illustrate the parameters sampled throughout world model training for the Terrain Walker and Clean Up domains.

Domain randomisation (DR) samples parameters uniformly throughout training. WAKER-R biases sampling towards environments where the uncertainty estimate is decreasing the fastest. For Terrain Walker, we see in Figure 7 that initially (0 - 2M steps), WAKER-R samples all parameters equally in a similar fashion to DR. This indicates that during this period, the uncertainty estimate for all environment parameters is decreasing approximately equally. From 2M - 5M steps, WAKER-R selects terrain with a higher amplitude, and shorter length scale (more complex and undulating terrain). This indicates that during this period, the uncertainty for the complex terrain is decreasing more quickly and therefore is sampled more often. Thus, we observe that for Terrain Walker WAKER-R initially focuses equally on all environments, before switching to sampling more complex terrain when simple domains have converged and therefore no longer exhibit a gradient in uncertainty.

WAKER-M biases sampling towards the domains with the highest magnitude of uncertainty. We do not expect WAKER-M to initially focus on the simplest environments, as it is always biased towards sampling complex environments where uncertainty is high. In Figure 7, we see that for Terrain Walker WAKER-M initially (0 - 0.3M steps) samples the parameters similar to DR. From 0.3M steps onwards, WAKER-M more frequently samples complex terrain with high amplitude and short length scale. Thus, WAKER-M consistently samples complex environments with higher uncertainty. Likewise, in Figure 8 we observe that for Clean Up WAKER-M more frequently samples environments with a larger arena size and more blocks. Thus, in Clean Up WAKER-M also consistently samples complex environments with higher uncertainty.

For WAKER-R in Clean Up, we observe that in the early stages (0-1M steps) sampling is heavily focused on the most complex environments with larger arenas and more blocks. This indicates that uncertainty is being reduced more quickly on the more complex environments during this stage of training. As training progresses, WAKER-R gradually places less emphasis on the most complex environments, and samples the environments more evenly.

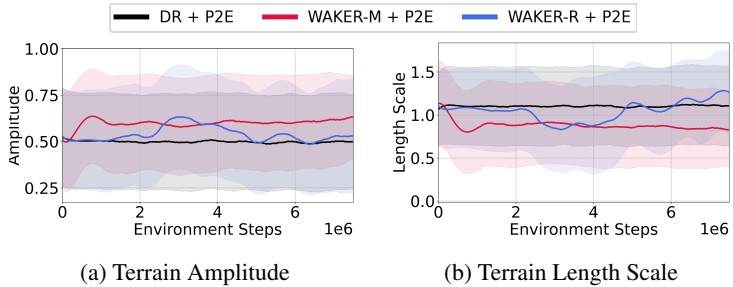

(a) Terrain Amplitude         (b) Terrain Length Scale

Figure 7: Plots of the parameters sampled for Terrain Walker domain.

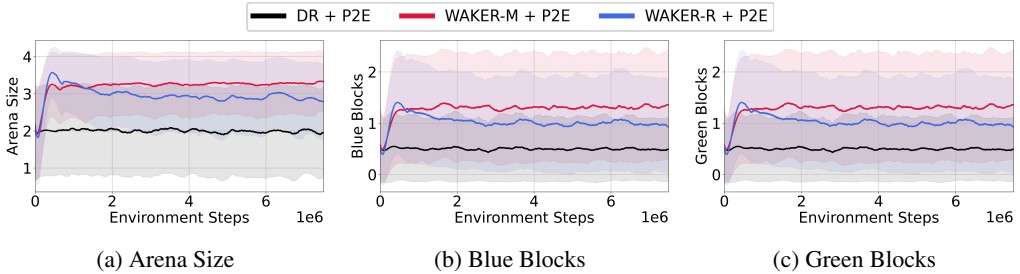

(a) Arena Size      (b) Blue Blocks      (c) Green Blocks

Figure 8: Plots of the parameters sampled for Clean Up domain.

## C.4    AVERAGE PERFORMANCE RESULTS

In Table 1 in the main results in Section 4, we presented the results for the robustness evaluation, where we averaged performance over the 10 worst runs on 100 randomly sampled environments. In Table 5 and Figure 9 we present the results from averaging over all 100 episodes. Note that we do not necessarily expect WAKER to improve over DR for this evaluation, as the DR baseline directly optimises for this exact training distribution.

Figure 9 shows that WAKER-M achieves a statistically significant improvement in average performance over DR when using either exploration policy. For WAKER-R, the average performance is not significantly different when compared to DR for either exploration policy.

| Exploration Policy | | **Plan2Explore** | | | | | **Random Exploration** | | |
|---|---|---|---|---|---|---|---|---|---|
| Environment Sampling | | **WAKER-M** | **WAKER-R** | **DR** | **GE** | **HE-Oracle** | **RW-Oracle** | **WAKER-M** | **WAKER-R** | **DR** |
| | Sort | $0.973 \pm 0.01$ | $0.966 \pm 0.01$ | $0.929 \pm 0.02$ | $0.932 \pm 0.01$ | $0.884 \pm 0.04$ | $0.942 \pm 0.02$ | $0.779 \pm 0.04$ | $0.766 \pm 0.03$ | $0.743 \pm 0.03$ |
| Clean Up | Sort-Rev. | $0.976 \pm 0.01$ | $0.957 \pm 0.01$ | $0.934 \pm 0.01$ | $0.949 \pm 0.01$ | $0.881 \pm 0.03$ | $0.960 \pm 0.02$ | $0.891 \pm 0.03$ | $0.958 \pm 0.02$ | $0.739 \pm 0.04$ |
| | Push | $0.972 \pm 0.01$ | $0.969 \pm 0.01$ | $0.957 \pm 0.01$ | $0.960 \pm 0.02$ | $0.891 \pm 0.03$ | $0.958 \pm 0.02$ | $0.839 \pm 0.03$ | $0.819 \pm 0.03$ | $0.786 \pm 0.04$ |
| | Sort | $0.988 \pm 0.01$ | $0.980 \pm 0.01$ | $0.967 \pm 0.01$ | $0.964 \pm 0.01$ | $0.816 \pm 0.05$ | $0.964 \pm 0.01$ | $0.936 \pm 0.02$ | $0.922 \pm 0.02$ | $0.909 \pm 0.01$ |
| Car Clean Up | Sort-Rev. | $0.991 \pm 0.01$ | $0.988 \pm 0.01$ | $0.963 \pm 0.02$ | $0.965 \pm 0.01$ | $0.828 \pm 0.04$ | $0.959 \pm 0.02$ | $0.918 \pm 0.02$ | $0.918 \pm 0.02$ | $0.896 \pm 0.02$ |
| | Push | $0.990 \pm 0.01$ | $0.989 \pm 0.01$ | $0.975 \pm 0.01$ | $0.981 \pm 0.01$ | $0.812 \pm 0.04$ | $0.981 \pm 0.01$ | $0.953 \pm 0.01$ | $0.939 \pm 0.02$ | $0.909 \pm 0.03$ |
| | Walk | $909.1 \pm 12.1$ | $915.4 \pm 10.3$ | $900.4 \pm 22.7$ | $905.8 \pm 13.9$ | $652.9 \pm 73.5$ | $873.7 \pm 26.6$ | $385.8 \pm 34.3$ | $360.5 \pm 42.0$ | $386.7 \pm 24.6$ |
| | Run | $393.3 \pm 8.5$ | $388.7 \pm 14.7$ | $397.5 \pm 13.1$ | $403.2 \pm 13.0$ | $261.9 \pm 21.4$ | $365.4 \pm 18.4$ | $149.4 \pm 10.7$ | $138.4 \pm 9.6$ | $149.6 \pm 15.2$ |
| Terrain Walker | Flip | $973.2 \pm 9.8$ | $965.3 \pm 8.5$ | $967.1 \pm 5.1$ | $973.9 \pm 7.1$ | $982.3 \pm 0.9$ | $973.7 \pm 7.4$ | $935.6 \pm 8.0$ | $924.3 \pm 13.7$ | $924.8 \pm 9.4$ |
| | Stand | $970.4 \pm 4.9$ | $973.3 \pm 6.8$ | $970.3 \pm 8.2$ | $968.0 \pm 9.0$ | $927.2 \pm 25.4$ | $962.4 \pm 12.8$ | $673.1 \pm 22.5$ | $669.6 \pm 66.7$ | $689.7 \pm 84.7$ |
| | Walk-Back. | $861.2 \pm 10.0$ | $845.0 \pm 25.1$ | $871.6 \pm 13.6$ | $848.0 \pm 14.8$ | $573.2 \pm 36.1$ | $808.1 \pm 18.2$ | $474.1 \pm 9.6$ | $450.9 \pm 19.2$ | $446.7 \pm 32.1$ |
| | Hop | $483.8 \pm 15.9$ | $483.9 \pm 14.8$ | $491.8 \pm 12.7$ | $490.8 \pm 10.5$ | $300.2 \pm 21.1$ | $456.5 \pm 15.2$ | $35.6 \pm 11.0$ | $27.1 \pm 12.7$ | $45.5 \pm 20.7$ |
| Terrain Hopper | Hop-Back. | $472.2 \pm 4.4$ | $470.4 \pm 16.8$ | $477.9 \pm 17.7$ | $480.8 \pm 8.9$ | $290.3 \pm 21.7$ | $456.0 \pm 10.2$ | $29.4 \pm 13.2$ | $19.0 \pm 13.3$ | $38.7 \pm 25.2$ |
| | Stand | $729.0 \pm 73.6$ | $811.0 \pm 31.6$ | $794.4 \pm 16.0$ | $790.3 \pm 41.6$ | $680.5 \pm 64.1$ | $771.5 \pm 37.0$ | $48.1 \pm 22.1$ | $53.3 \pm 30.5$ | $49.6 \pm 14.9$ |

Table 5: Average performance evaluation: Average performance on 100 randomly sampled environments. For each exploration policy, we highlight results within 2% of the best score, and $\pm$ indicates the S.D. over 5 seeds. For Terrain Hopper with a Random Exploration policy, all methods fail to learn a meaningful policy so none are highlighted.

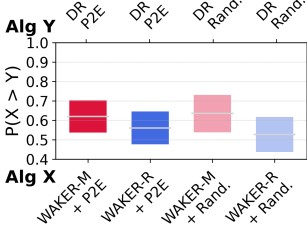

Figure 9: Average performance aggregated CIs.

# D    FURTHER RESULTS

Here, we present additional results that expand upon the results already presented in the main paper.

## D.1    ADDITIONAL WORLD MODEL IMAGE PREDICTION ERRORS

To compare the quality of the world model predictions, we compute the errors between the images predicted by the world model, and the actual next image. For each world model, we collect 200 trajectories in each of 200 randomly sampled environments. For the results in the main paper (Figure 6), the trajectories are collected under performant task policies (for the *hop* task for Terrain Hopper, and for the *sort* task for Car Clean Up). For the results in Figures 10 and 11 the trajectories are collected under a uniform random policy.

Along each trajectory, we compute the next image predicted by the world model by decoding the mean of the next latent state prediction, using the decoder learned by DreamerV2. We compute the mean squared error between the image prediction and the actual next image. Then, we average the errors along the trajectory to compute the error for the trajectory. We repeat this process along all trajectories to compute the error for each trajectory. Then, we compute $\text{CVaR}_{0.1}$ of these error evaluations by taking the average over the worst 10% of trajectories. These values are plotted for each domain in Figures 6, 10, and 11.

We observe that the CVaR of the image prediction errors are generally lower for world models trained with WAKER than DR. The difference in performance is especially large for the Clean Up and Car Clean Up domains. This mirrors the large difference in policy performance between WAKER and DR on the Car Clean Up and Clean Up domains. This verifies that WAKER is successfully able to learn world models that are more robust to different environments than DR. For Terrain Walker and Terrain Hopper, the difference in the error evaluations between methods is smaller, but we still observe that WAKER produces smaller errors than DR when using the Plan2Explore exploration policy.

Note that in Figures 10 and 11 the error evaluation is performed using trajectories generated by a uniform random policy. Therefore, we observe that the world models trained using data collected by a random policy tend to have lower errors relative to world models trained using Plan2Explore as the exploration policy.

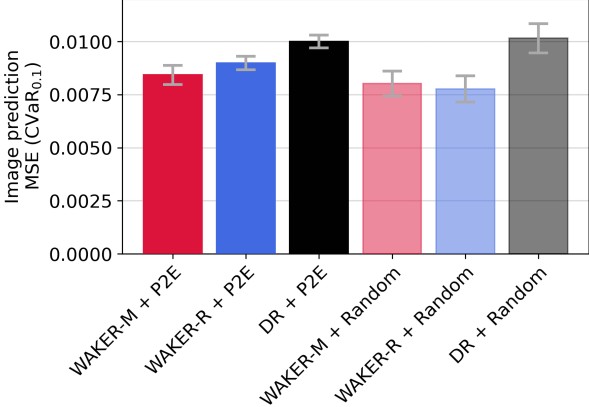

Figure 10: Clean Up image prediction errors for a uniform random policy. Error bars indicate standard deviations across 5 seeds.

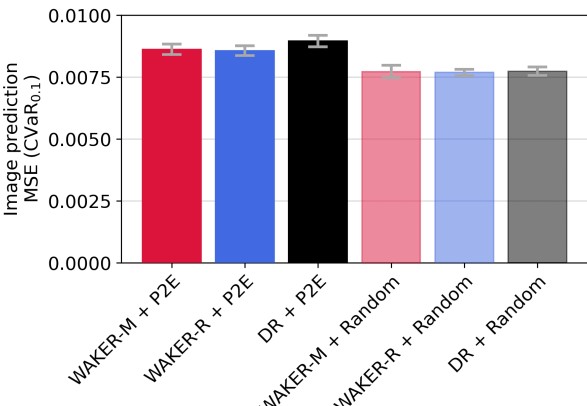

Figure 11: Terrain Walker image prediction errors for a uniform random policy. Error bars indicate standard deviations across 5 seeds.

## D.2 PERFORMANCE ON SNAPSHOTS OF WORLD MODEL TRAINED WITH VARIABLE AMOUNTS OF DATA

In the main results presented in Section 4, we report the performance of policies obtained from the *final* world model at the end of six days of training. In this section, we present the performance of policies obtained from snapshots of the world model trained with variable amounts of data. These results are presented in Figures 12, 13 and 14. In each of the plots, the $x$-axis indicates the number of steps of data collected within the environment that was used to train the world model. The values on the $y$-axis indicate the performance of policies obtained from imagined training within each snapshot of the world model.

As expected, we observe that as the amount of data used to train the world model increases, the performance of policies trained within the world model increases. We observe that for most levels of world model training data, and for both exploration policies, the WAKER variants improve upon DR in the robustness and OOD evaluations in Figures 12 and 13. The magnitude of the improvement tends to become larger as the amount of training data from the environment increases. This suggests that WAKER might be especially effective when scaled to larger amounts of training data.

The average performance of policies trained in snapshots of the world model is presented in Figure 14. For Terrain Walker and Terrain Hopper, the average performance between WAKER and DR for the same exploration policy is similar across all levels of environment data. For Clean Up and Car Cleanup, we observe that both WAKER variants improve the average performance relative to DR when using Plan2Explore as the exploration policy, for almost all levels of data. For the random exploration policy, we observe that the average performance initially improves more slowly when using WAKER, but eventually surpasses the performance of DR. This initially slower improvement in average performance with WAKER is likely due to WAKER putting increased emphasis on collecting data from more complex environments.

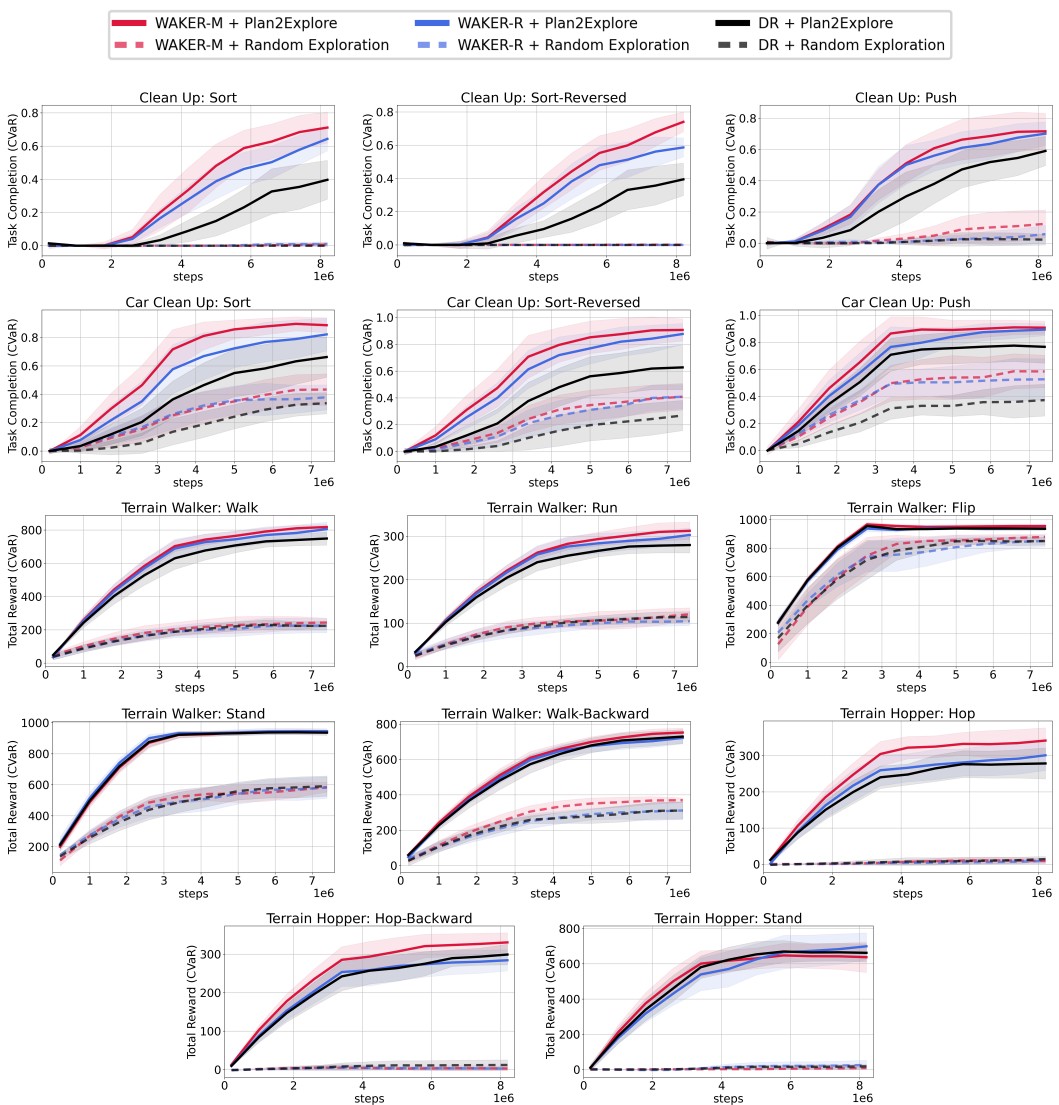

Figure 12: Robustness evaluation. Plots show the $CVaR_{0.1}$ metric: the mean of worst 10 runs on 100 randomly sampled environments. The policies are trained zero-shot on snapshots of world model trained on amount of data labelled on $x$-axis. Results are averaged across 5 seeds, and shaded regions are standard deviations across seeds.

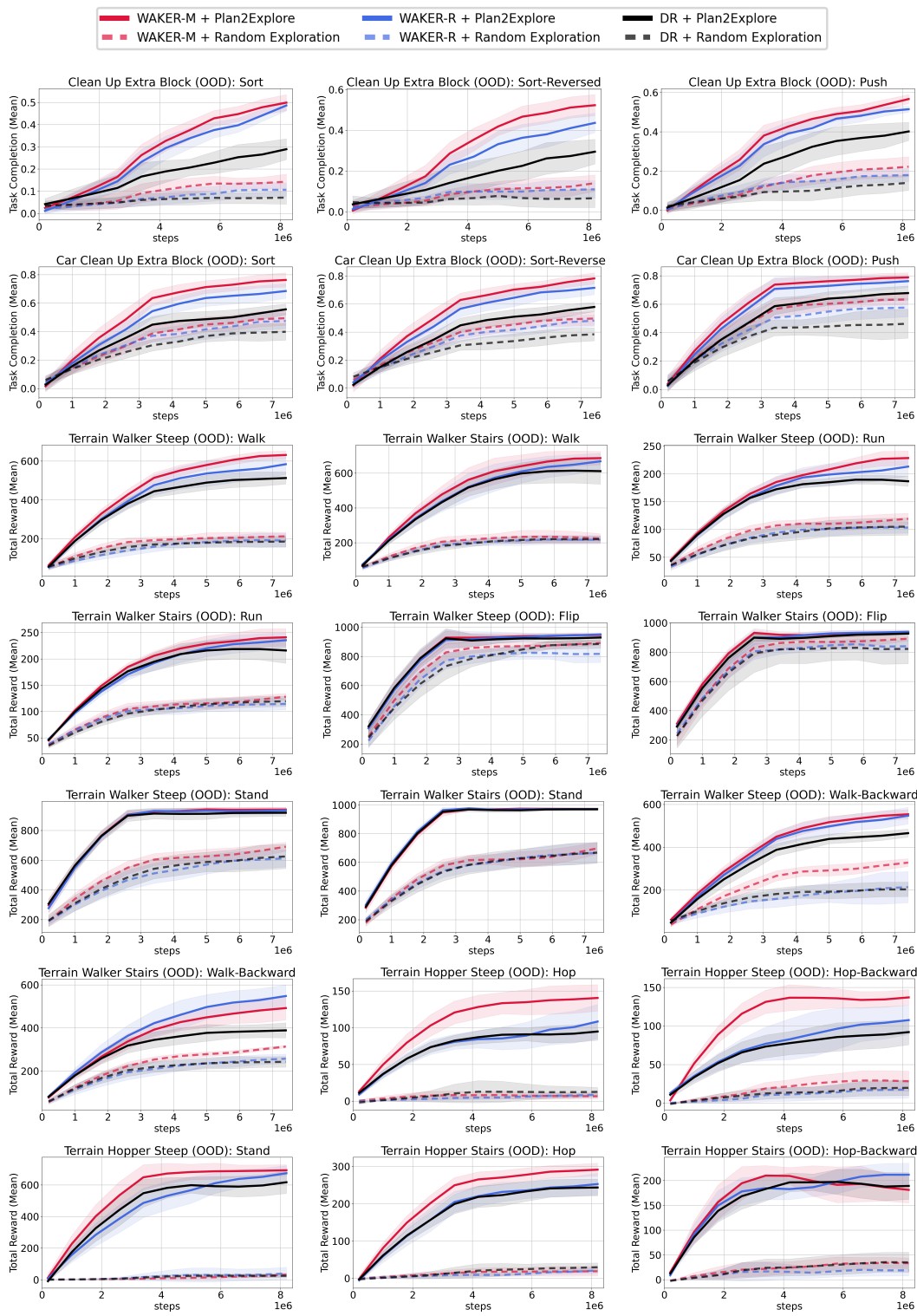

Figure 13: Out-of-distribution evaluation. Plots show the average performance on out-of-distribution environments. The policies are trained zero-shot on snapshots of world model trained on amount of data labelled on $x$-axis. Results are averaged across 5 seeds, and shaded regions are standard deviations across seeds.

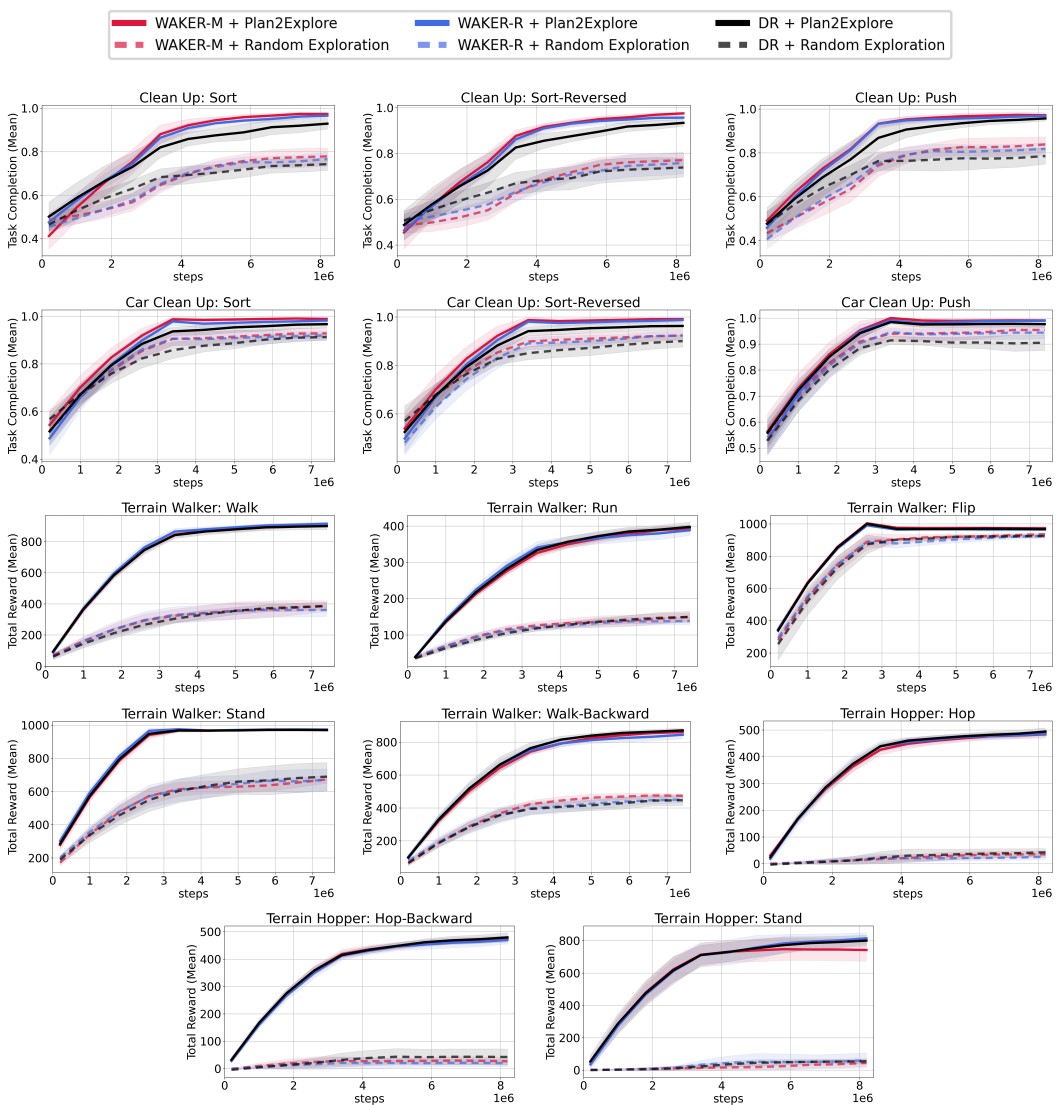

Figure 14: Average performance. Plots show the average performance on 100 evaluations on environments sampled uniformly at random. The policies are trained zero-shot on snapshots of the world model trained on amount of data labelled on $x$-axis. Results are averaged across 5 seeds, and shaded regions are standard deviations across seeds.

# E  WAKER IMPLEMENTATION DETAILS

Here, we provide additional details on our implementation of the WAKER algorithm that were omitted from the main paper due to space constraints. The code for our experiments is available at github.com/marc-rigter/waker.

**Error Estimate Update and Smoothing**  In Line 17 of Algorithm 1 we compute the error estimate $\delta_\theta$ according to the average ensemble disagreement over an imagined trajectory. Each $\delta_\theta$ value is noisy, so we maintain a smoothed average $\overline{\delta}_\theta$ of the $\delta_\theta$ values. Specifically, we use an exponential moving average (EMA). For each parameter setting $\theta$, when we receive a new $\delta_\theta$ value, we update the EMA according to: $\overline{\delta}_\theta \leftarrow \alpha \overline{\delta}_\theta + (1 - \alpha)\delta_\theta$, where we set $\alpha = 0.9999$.

For WAKER-M, the error buffer $D_{\text{error}}$ contains the smoothed average of the $\delta_\theta$ values for each parameter setting: $D_{\text{error}} = \{\overline{\delta}_{\theta_1}, \overline{\delta}_{\theta_2}, \overline{\delta}_{\theta_3}, \ldots\}$. These values are used as input to the Boltzmann distribution used to sample the environment parameters in Line 7 of Algorithm 1.

For WAKER-R, we compute the reduction of the $\overline{\delta}_\theta$ values for each parameter setting between each interval of 10,000 environment steps: $\Delta_\theta = \overline{\delta}_\theta^{\text{old}} - \overline{\delta}_\theta^{\text{new}}$. Because the error estimates change very slowly, we perform a further smoothing of the $\Delta_\theta$ values using an exponential moving average: $\overline{\Delta}_\theta \leftarrow \alpha_\Delta \overline{\Delta}_\theta + (1 - \alpha_\Delta)\Delta_\theta$, where we set $\alpha_\Delta = 0.95$. For WAKER-R, the error estimate buffer contains these $\overline{\Delta}_\theta$ values: $D_{\text{error}} = \{\overline{\Delta}_{\theta_1}, \overline{\Delta}_{\theta_2}, \overline{\Delta}_{\theta_3}, \ldots\}$, and these values determine the environment sampling distribution in Line 7 of Algorithm 1.

**Error Estimate Normalisation**  We normalize the error values in Line 7 of Algorithm 1 to reduce the sensitivity of our approach to the scale of the error estimates, and reduce the need for hyperparameter tuning between domains. For WAKER-R, we divide each $\overline{\Delta}_\theta$ value by the mean absolute value of $\overline{\Delta}_\theta$ across all parameter settings. The rationale for this form of normalisation is that if the rate of reduction of error is similar across all environments, then WAKER-R will sample the environments with approximately equal probability.

For WAKER-M, for each $\overline{\delta}_\theta$ value, we subtract the mean of $\overline{\delta}_\theta$ across all parameter settings, and divide by the standard deviation. This means that regardless of the scale of the error estimates, WAKER-M will always favour the environments with the highest error estimates, as motivated by Problem 2.

**World Model Training**  For the world model, we use the official open-source implementation of DreamerV2 (Hafner et al., 2021) at `https://github.com/danijar/dreamerv2`. For the world model training we use the default hyperparameters from DreamerV2, with the default batch size of 16 trajectories with 50 steps each. For the ensemble of latent dynamics functions, we use 10 fully-connected neural networks, following the implementation of Plan2Explore (Sekar et al., 2020) in the official DreamerV2 repository. We perform one update for every eight environment steps added to the data buffer.

In our implementation of Plan2Explore, each member of the ensemble is trained to predict both the deterministic and stochastic part of the next latent state. This is slightly different to the Plan2Explore implementation in the official Dreamer-v2 codebase, where the ensemble only predicts part of the latent state. We made this modification because the derivation of our algorithm indicates that we should be concerned with the uncertainty over the prediction of the entire latent state.

**Policy Training**  For the exploration and task policies, we use the actor-critic implementation from the official DreamerV2 repository. During each update, we sample a batch of 16 trajectories from the data buffer of 50 steps each. For each latent state corresponding to a data sample in the batch, we generate an imagined trajectory starting from that latent state, with a horizon of 15 steps. We then update both the task and exploration policies using dynamics gradients (i.e. backpropagation through the world model) computed on these imagined trajectories. We perform one update to each of the task and exploration policies for every eight environment steps.

# F EXPERIMENT DETAILS

## F.1 DOMAINS AND TASKS

**Terrain Walker and Terrain Hopper**   We simulate the Walker and Hopper robots from the Deep-Mind Control Suite (Tassa et al., 2018). The observations are images of shape $64 \times 64 \times 3$. For each episode, we generate terrain using Perlin noise (Perlin, 2002), a standard technique for procedural terrain generation. The terrain generation is controlled by two parameters, the amplitude, with values in $[0, 1]$, and the length scale, with values in $[0.2, 2]$. Domain randomisation samples both of these parameters uniformly from their respective ranges. For examples of each domain and the terrain generated, see Figures 15 and 16.

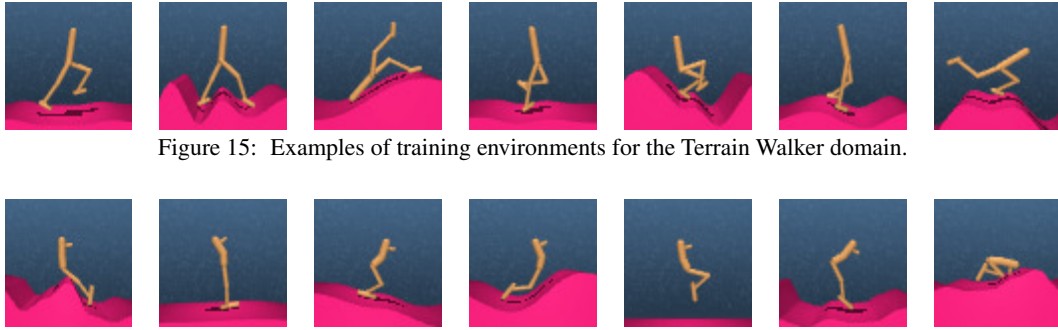

Figure 15: Examples of training environments for the Terrain Walker domain.

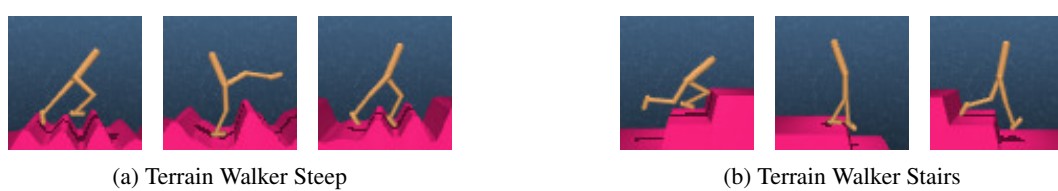

Figure 16: Examples of training environments for the Terrain Hopper domain.

We evaluate 5 different downstream tasks for Terrain Walker. The tasks *walk*, *run*, *stand*, *flip* are from the URLB benchmark (Laskin et al., 2021). We also include *walk-backward*, which uses the same reward function as *walk*, except that the robot is rewarded for moving backwards rather than forwards.

We evaluate 3 different downstream tasks for Terrain Hopper: *hop*, *stand* and *hop-backward*. *Hop* and stand are from the Deepmind Control Suite Tassa et al. (2018). *Hop-backward* is the same as *hop*, except that the robot is rewarded for moving backwards rather than forwards.

**Terrain Walker and Terrain Hopper: Out-of-Distribution Environments**   We use two different types of out-of-distribution environments for the terrain environments. The first is *Steep*, where the length scale of the terrain is 0.15. This is 25% shorter than ever seen during training. This results in terrain with sharper peaks than seen in training. The second is *Stairs*, where the terrain contains stairs, in contrast to the undulating terrain seen in training. The out-of-distribution environments are shown in Figure 17 for the Walker robot. The out-of-distribution environments for Terrain Hopper are the same, except that we use the Hopper robot.

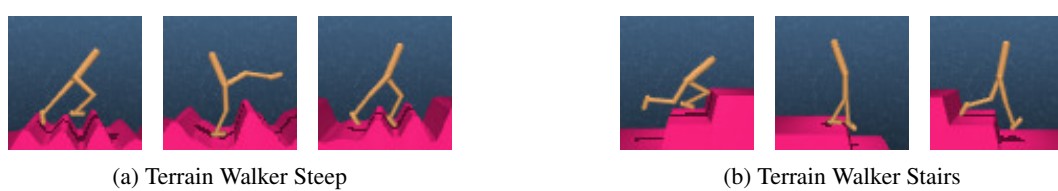

(a) Terrain Walker Steep                    (b) Terrain Walker Stairs

Figure 17: Examples of out-of-distribution (OOD) environments for the Terrain Walker domain. The OOD environments for Terrain Hopper are the same, except that we use the Hopper robot.

**Clean Up and Car Clean Up**   These domains are based on SafetyGym (Ray et al., 2019) and consists of a robot and blocks that can be pushed. The observations are images from a top-down view with dimension $64 \times 64 \times 3$. For Clean Up, the robot is a point mass robot. For Car Clean Up the robot is a differential drive car. In both cases, the action space is two-dimensional.

For each environment, there are three factors that vary: the size of the environment, the number of blocks, and the colour of the blocks. For the default domain randomisation sampling distribution, the size of the environment is first sampled uniformly from size $\in \{0, 1, 2, 3, 4\}$. The number of

blocks is then sampled uniformly from $\{0, 1, \ldots, \texttt{size}\}$. The number of green vs blue blocks is then also sampled uniformly. Examples of the training environments generated for the Clean Up domain are in Figure 18.

There are three different tasks for both Clean Up and Car Clean Up: *sort*, *push*, and *sort-reversed*. The tasks vary by the goal location for each colour of block. For *sort*, each block must be moved to the goal location of the corresponding colour. For *sort-reverse*, each block must be moved to the goal location of the opposite colour. For *push*, all blocks must be pushed to the blue goal location, irrespective of the colour of the block.

We define the *task completion* to be the number of blocks in the environment that are in the correct goal region divided by the number of blocks in the environment. If there are no blocks in the environment, then the task completion is 1. The reward function for each task is defined as follows: The agent receives a dense reward for moving any block closer to the desired goal region, and the agent also receives a reward at each time step that is proportional to the task completion.

In the main results we report the task completion at the end of each episode, as it is easier to interpret. We observe that the results for the total reward directly correlate to those for the task completion.

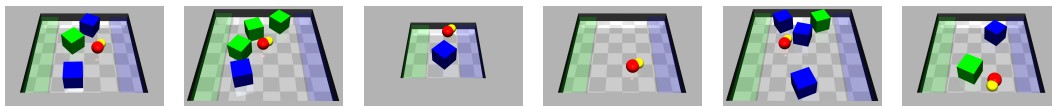

Figure 18: Examples of training environments for the Clean Up domain. The environments differ by their size, the number of blocks, and the colour of each block. The shaded regions indicate the goal locations.

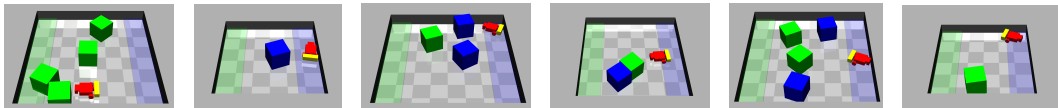

Figure 19: Examples of training environments for the Car Clean Up domain.

**Clean Up and Car Clean Up: Out-of-Distribution Environments**  For the out-of-distribution environments, we place one more block in the environment than was ever seen during training. We set the size of the environment to $\texttt{size} = 4$, and there are 5 blocks in the environment. The task completion and reward function is defined the same as for the training environments. Examples of the out-of-distribution environments are in Figure 20.

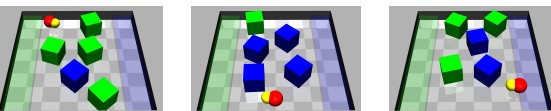

Figure 20: Examples of out-of-distribution (OOD) environments for Clean Up Extra Block domain. For Car Clean Up, the OOD environments are the same except that we use the car robot.

### F.2  LENGTH OF TRAINING RUNS

Due to resource limitations, each training run is limited to six days of run time. This corresponds to a total of $7.4 \times 10^6$ environment steps for the Terrain Walker and Terrain Hopper domains, and a total of $8.2 \times 10^6$ environment steps for the Clean Up and Car Clean up domains. The results reported in Section 4 are for task policies trained in imagination in the final world models at the end of these training runs.

### F.3  HYPERPARAMETER TUNING

We use the default parameters for DreamerV2 (Hafner et al., 2021) for training the world model.

For WAKER, there are two hyperparameters: the probability of sampling uniformly from the default environment distribution, $p_{\text{DR}}$, and the temperature parameter for the Boltzmann environment distribution, $\eta$. In our experiments, we set $p_{\text{DR}} = 0.2$ for all experiments and did not tune this value. We performed limited hyperparameter tuning of the Boltzmann temperature parameter, $\eta$. We ran

WAKER-M + Plan2Explore and WAKER-R + Plan2Explore for each of three different values of $\eta \in \{0.5, 1.0, 1.5\}$. For each algorithm and $\eta$ value, we ran two seeds for 5e6 environment steps on the Clean Up domain. At the end of 5e6 environment steps, we chose the value of $\eta$ that obtained the best performance for each algorithm for $CVaR_{0.1}$ on the *sort* task. We then use this value of $\eta$ for WAKER-M and WAKER-R across all experiments, when using both the Plan2Explore and the random exploration policies.

The hyperparameters used in our experiments for WAKER are summarised in Table 6.

Table 6: Summary of hyperparameters used by WAKER.

|  | WAKER-M | WAKER-R |
|---|---|---|
| $p_{DR}$ | 0.2 | 0.2 |
| $\eta$ | 1 | 0.5 |

### F.4 Confidence Interval Details

Figures 3, 4 and 9 present 95% confidence intervals of the probability of improvement, computed using the *rliable* framework (Agarwal et al., 2021). To compute these values, we first normalise the results for each algorithm and task to between $[0, 1]$ by dividing by the highest value obtained by any algorithm. We then input the normalised scores in the *rliable* package to compute the confidence intervals.

A confidence interval where the lower bound on the probability of improvement is greater than 0.5 indicates that the algorithm is a statistically significant improvement over the baseline.

### F.5 Computational Resources

Each world model training run takes 6 days on an NVIDIA V100 GPU. In our experiments, we train 120 world models in total, resulting in our experiments using approximately 720 GPU days.

## G Baseline Method Details

In our experiments, we compare the following methods. Note that two of the baselines (HE-Oracle and RW-Oracle) require expert domain knowledge. The other methods (WAKER, GE, and DR) do not require domain knowledge.

**Domain Randomisation (DR)** DR samples environments uniformly from the default environment distribution (as described in Appendix F.1).

**Hardest Environment Oracle (HE-Oracle)** For this baseline, the most complex instance of the environment is always sampled. For the block pushing tasks, the most complex environment is the largest arena, containing two blocks of each colour. For the terrain tasks, the most complex environment is the terrain with the highest possible amplitude (1) and the shortest possible length scale (0.2).

**Re-weighting Oracle (RW-Oracle)** RW-Oracle re-weights the environment distribution to focus predominantly on the most complex environments. 20% of the time RW-Oracle samples from the default domain randomisation distribution. The remaining 80% of the time RW-Oracle samples uniformly from the most complex environments. For the terrain environments, the most complex environments are those where the amplitude is within $[0.8, 1]$ and the length scale is within $[0.2, 0.4]$. For the block pushing environments, the most complex environments are those where the size of the arena is four and there are four blocks of any colour.

**Gradual Expansion (GE)** 20% of the time, a new environment is sampled from the default domain randomisation distribution. The remaining 80% of the time, GE samples from the default distribution, but the default distribution is restricted to only include environments that have been seen so far. Thus, GE utilises the default domain randomisation distribution to sample new environments to gradually increase the range of environments seen during training.

