# OpenReview forum: "Reward-Free Curricula for Training Robust World Models"
_ICLR.cc/2024/Conference — ICLR 2024 poster_

### Official Review · Reviewer_3ipu · 2023-10-26

**Soundness:** 2 fair
**Presentation:** 3 good
**Contribution:** 4 excellent
**Rating:** 6
**Confidence:** 4

**Summary:**

This work addresses the challenge of learning a robust World Model in a task-agnostic setting, i.e., without the supervision of any reward distribution. For this matter, it formulates the problem as a minimax regret objective and proves that, under assumptions of learnability of the optimal policy (under reward supervision) and existence of a latent dynamics that models the real environment, the regret of a optimal “World Model” policy is bounded and depends on the state-action distribution and the total variation between the learnt latent and the true dynamics. Based on that, the paper introduces WAKER, an agent that actively samples environments to generate a reward-free curricula in order to learn a robust World Model. Experiments show that WAKER outperforms domain randomization in robustness and out-of-distribution settings.

**Strengths:**

- The work introduces the setting of generating task-agnostic curricula for learning robust world models, arguing that this direction may help learning generally-capable agents, which is an interesting perspective with good motivation. Furthermore, the paper is well written and didactic.
- The paper mathematically formalizes learning robust World Models under a framework of minimax regret and proves a regret bound for optimal policies under learned World Models. For this, it clearly states the assumptions and presents the proof. The theoretical development is sound and clear.
    - Furthermore, the derivation starting from the theoretical framework and arriving into the proposed learning agent (WAKER) is clever and well presented.

- The experimental setting shows the effectiveness of WAKER in robust and OOD evaluation settings in comparison with the random sampling approach (DR). It also ablates the exploration policy and shows an interesting and interpretable illustration of the generated curricula.

**Weaknesses:**

- The proposed learning objective (and, therefore, learning agent) relies on a strong assumption that seems to be overlooked (or at least not discussed with appropriate depth): the “capacity” of the exploration policy to cover the full state-action distribution. Equivalently, the relaxation of the bound by introducing an exploration policy is a big assumption and limits the generality of the proposed agent. It is questionable whether a single intrinsically-motivated policy could explore regions of the state-action space associated with high rewards for locomotion gaits (as in the walker scenario) with terrain variability, for instance.

- Regarding “Zero-Shot Task Adaptation”: The presented methodology for learning task-specific policies relies on the acquired dataset “D” to train the reward predictor. Nevertheless, this raises a similar problem to the previous concern: if D does not contain the state-action pairs associated with high rewards for a particular task, then the learned policy might be potentially very suboptimal. If the work relies on Assumption 1 to disregard this problem, then it should be clearly stated in the paper, but I believe that this would make Assumption 1 unreasonable.

- In Figure 6 and 10, I understood that the prediction error is computed as the mean squared error on the pixel values. If this is indeed the case, it is questionable if this is the best metric to evaluate the prediction error. Furthermore, collecting the test set by sampling a random policy will omit part of the state space, and likely very relevant states (i.e., associated with high rewards in an optimal policy for locomotion). Therefore, the results are not quite convincing. I recommend trying to use the underlying physics of the simulator as a “true latent” dynamics and compute the distance from it.  Another possibility is to learn a separated embedding space (via metric learning or autoencoding) and use it as the ground truth representation. Additionally, I also recommend learning optimal policies for several different environments and use the history of policies generated during training to sample from the space. This would give a more complete covering of the state-action distribution.


**Summary of the Review**:

Overall, I believe the paper provides solid contributions, for introducing a new setting, and also providing a sound theoretical framework and cleverly translates it to a learning algorithm. Nevertheless, the introduction setting must consider the challengings of covering the state-action distribution correctly, which was overlooked in the paper. Furthermore, the experiments regarding model prediction error can be improved to be convincing.  From my perspective, concerns 1 and 2 should be addressed by clarifying these limitations in the paper. Concern 3 can be addressed by the suggestions in the experimental setting. *I will recommend acceptance if these concerns were addressed.*

**Questions:**

- Following my second concern, does Assumption 1 imply that the learned latent dynamics will have support in the full state-action distribution?

- Following my third concern, is the image prediction error indeed computed on pixel-based mean-squared error?


====================== **POST-REBUTTAL** =======================

I would like to thank the authors for putting efforts on the rebuttal discussion and addressing concerns. I believe my concerns were partially addressed, and, as a consequence, I am leaning towards acceptance of the paper (5 -> 6). Overall, as described in my initial review, my main concerns are related to the assumptions that the work is built upon. Although I still believe they are strong and limit the applicability of the method in harder scenarios, I also understand that these assumptions are present in previous work and the authors updated the work to highlight these as potential limitations, improving transparency.

In more details:

- For my first point: the authors addressed the concern as possible, and showed average returns to provide evidence that, for the particular tasks in the paper, this was not a problem.

- For my second point: similarly, they provided discussion on the limitations and clarified my question.

- For my third point: I think it was partially addressed. The paper now provides evidence under a performant policy (rather than random policy) which substantially supports the validity of the experiments. I still have my concerns of using MSE as a predictive metric (and how much semantic value it brings). Nevertheless, I understand that it would indeed require an inverse mapping between image and true physics simulator state, which does not seem easy to obtain as mentioned.

---

> ### Author Response · Authors · 2023-11-19
> **Response to Reviewer 3ipu**
>
> Thank you for your review!  We have added results for additional baselines as described in the general response. We will respond to each of your comments below.
>
> &nbsp;
> > World Model Error Prediction
>
> Yes, our evaluation of world model error is computed in terms of the mean-squared error of pixel-based predictions. **We have revised the error results in the main body of the paper so that they are evaluated under a performant task-specific policy** (rather than a random policy) as you have suggested. The new results for error predictions are qualitatively similar to our original results. The original error results under a uniform random policy are deferred to the appendix.
>
> You have also suggested that we should compare the accuracy against the underlying physics simulator state. However, the world model only receives image observations as input, and decodes images. Therefore, it does not make any predictions about the physics state of the simulator. So we are not able to make this suggested evaluation.
>
> &nbsp;
> > It is questionable whether a single intrinsically-motivated policy could explore regions of the state-action space associated with high rewards
>
> As you point out, our work assumes that an intrinsically motivated agent will sufficiently explore regions of the state-action space that are relevant to learning performant policies for downstream tasks. **We have added discussion of this assumption and potential limitation to the conclusion.** The conclusion now reads:
>
> *In future work, we are excited to scale our approach to even more complex domains with many variable parameters. One limitation of our approach is that it relies upon an intrinsically motivated policy to adequately explore the state-action space across a range of environments. This may pose a challenge for scalability to more complex environments. To scale WAKER further, we plan to make use of function approximation to estimate uncertainty throughout large parameter spaces, as opposed to the discrete buffer used in this work. We also plan to investigate using WAKER for reward-free pretraining, followed by task-specific finetuning
> to overcome the challenge of relying upon intrinsically motivated exploration.*
>
> We would like to point on that this assumption is also made in other works that consider reward-free world model learning (such as Plan2Explore). As the results from the Plan2Explore paper demonstrate, it is possible to learn performant policies for downstream tasks such as locomotion by using only intrinsically motivated exploration.
>
> Likewise, the results from our paper show that it is possible to learn performant policies for a range of tasks *and environments* by only using intrinsically motivated exploration. The terrain locomotion results in our paper use the same reward functions as some of the DeepMind Control Suite (DMC) [1] tasks. The rewards received are a similar order of magnitude as those obtained by online RL algorithms in DMC [1], despite the fact that we use only intrinsically motivated exploration and train and evaluate across varying terrain. In the block pushing tasks in our paper, the best learned policies achieve > 95% in terms of average completion (see Table 5 in Appendix B.4), which also demonstrates that the policies learned from intrinsically motivated exploration are performant across a range of environments.
>
> [1] Tassa, Yuval, et al. "Deepmind control suite." *arXiv preprint arXiv:1801.00690* (2018).
>
> &nbsp;
> > Learning the reward function from the dataset
>
> We agree that the dataset must cover high-reward regions to learn an accurate reward function, and successfully learn a particular task. As mentioned above, we have added a discussion of the limitations of intrinsically motivated exploration to the conclusion.
>
> Assumption 1 states that we can recover the optimal policy *within the world model*. It makes no assumption that the world model (or the learnt reward function) is accurate throughout the state-action space. The assumption that addresses the coverage of the state-action space is that we can find a suitable $\pi_\theta^{expl}$ that will seek out the regions of the state-action space with high uncertainty. If the state-action space is very large, it may be infeasible for this exploration policy to explore all regions with high uncertainty.
>
> &nbsp;
> > Does Assumption 1 imply that the learned latent dynamics will have support in the full state-action distribution?
>
> The world model is defined and implemented in such a way that it will embed any sequence of previous observations and actions into the latent space. Then, conditioned on the current action, it will make a prediction over the next latent state. So the answer is that yes, it is the case that the learned latent dynamics will take any sequence of observations and actions and make a prediction for the next latent state (i.e. it has full support). Note, however, that Assumption 1 does not imply that the prediction will be accurate.

---

> > ### Author Response · Authors · 2023-11-22
> > **Follow Up**
> >
> > Dear Reviewer 3ipu,
> >
> > As the rebuttal period is drawing to a close, we wanted to check whether we have addressed your concerns. We would also like to point out that we have added extensive new results for new baselines. We believe that we have satisfied the changes that you requested, per our previous response. As a result of these changes (and the new baselines), we would appreciate it if you would consider raising your score to recommend acceptance, as you mentioned in your original review.
> >
> > Thank you for your time,
> >
> > The authors

---

> > > ### Comment · Reviewer_3ipu · 2023-11-22
> > > **Thanks for the clarifications**
> > >
> > > Dear authors,
> > >
> > > Thank you for your efforts during the rebuttal discussion. I updated my initial review the reflect that. Please check the Post-Rebuttal section. I am now leaning towards acceptance.

---

> > > > ### Author Response · Authors · 2023-11-22
> > > > **Thank you for your response**
> > > >
> > > > Fantastic, thank you for taking the time to read our response and get back to us! - The authors

---

### Official Review · Reviewer_hsNK · 2023-10-29

**Soundness:** 4 excellent
**Presentation:** 4 excellent
**Contribution:** 3 good
**Rating:** 8
**Confidence:** 4

**Summary:**

The work presents an approach for unsupervised environment design in a reward-free setting, using world models and based on the idea of minimax regret. By adopting WAKER, the agent samples environments based on the expected world model error, which is approximated using an ensemble of dynamics models.

**Strengths:**

* **Presentation**: the paper is well-written and the presentation is clear and clean. The metrics plotted in the Results section are sensible and the empirical results well reflect the story of the work.
* **Method**: the idea behind this work is innovative and well-motivated. Given the absence of previous work in this specific setting, I think the method will be useful for future work. The results also corroborate the usefulness of the approach.

**Weaknesses:**

* **Breaking assumptions in implementation**: I think the approach is well-motivated theoretically. However, in practice, the authors adopted a world model implementation that learns the dynamics using a recurrent module. By exploiting memory, the model is not encouraged to learn a Markovian latent state space. Given this is one of the assumptions behind the work, I think the authors should have opted for a memoryless model (e.g. learning from sequences of data but without keeping a memory) or they should make clear that their implementation is possibly breaking the assumptions (despite providing the expected improvements in practice)

**Questions:**

* In [1], the authors present the "latent dynamics discrepancy" metrics to evaluate the world model's misspecification when trained in reward-free settings and show how it correlates with downstream task performance. I think the idea behind this metric is relevant to the latent dynamics error used here and might be worth discussing in related work or exploring the way it is used in [1] for future work
* In Assumption 2, the authors talk about "modelling the real environment exactly", why what matters for the model is actually that the expected rewards are consistent along the distribution of state-actions visited by the actor. I suggest finding a better wording for that.
* The approach seems to work well in cases where the set of parameters $\theta$ for the environment's design is discrete/small. I recommend discussing how to extend the work for continuous sets of parameters, as this may require further approximations.

Typos/writing suggestions:
- in Preliminaries, there is a $\Delta(X)$ which I think should be $\Delta(S)$
- In section 3.1, "Futhermore"

[1] Rajeswar et al, 2023, Mastering the Unsupervised Reinforcement Learning Benchmark from Pixels

**Post rebuttal** : the authors have clarified their implementation choices and addressed my concerns. They also extended their evaluation, according to some other reviewers' suggestions. Now, I feel more confident about recommending acceptance for the paper and I have updated my score to reflect this.

---

> ### Author Response · Authors · 2023-11-19
> **Response to Reviewer hsNK**
>
> Thank you for your review!  We have added results for additional baselines as described in the general response. We will respond to your individual comments and questions below.
>
> &nbsp;
> ## Implementation and learning a Markovian latent space
>
> The **desire to learn a Markovian latent space is the reason why we chose to use a recurrent model** to compute the latent state. The intuition for why we can treat the recurrent latent state as Markovian is as follows. The RNN encodes the *history* of previous observations into the latent state. The history of observations is a sufficient statistic for the *belief* over partially observable variables [7]. It is well known that if we treat the belief as the state of an MDP, then the optimal behaviour in this *Markovian* model is the optimal behaviour for the original non-Markovian problem [8]. This idea is commonly referred to as the *belief MDP* [8]. Thus, by treating output of a recurrent module as the state, the resulting model is Markovian. Therefore, using a recurrent model allows us to fulfil our assumption of a Markovian latent space.
>
> This is a well-established assumption in many works on deep RL. Previous works on Deep RL have established that utilising an RNN enables non-Markovian problems (such as POMDPs or Meta RL) to be treated as Markovian [1, 2]. In the context of world models specifically, this assumption is also common. For example, the seminal paper [3] addresses POMDPs (i.e. non-Markovian problems) with a recurrent world model. The authors of [3] state that “The latent dynamics define a Markov decision process (MDP; Sutton, 1991) that is fully observed because the compact model states are Markovian”. The same assumption is also made in other previous works on world models such as [4, 5, 6].
>
> In light of this clarification, we would appreciate it if you would consider raising your score.
>
> &nbsp;
> ## Latent dynamics discrepancy
>
> Thank you for pointing out this relevant concept, which evaluates the difference between the latent predictions of a pretrained world model and a world model that is fine-tuned to a specific task. In our work, we do not finetune the world model to each specific task (we only pretrain the world model using reward-free exploration). Therefore, we cannot use this evaluation metric in our problem setting.
>
> We have added a reference to this work to the introduction as we agree it is relevant.
>
> &nbsp;
> ## Wording of Assumption 2
>
> We have changed the wording to say “for which the expected reward is the same as the real environment”.
>
> &nbsp;
> ## Future work: Large space of parameters
>
> If the space of parameters is much larger, we think that our work could be extended by using a function approximator to estimate the uncertainty across the space of parameters (rather than the discrete buffer of uncertainty estimates in our current implementation). We have added this discussion to the conclusion.
>
> &nbsp;
> ## Typos/Writing Suggestions
>
> Thank you for pointing these out, we have made these corrections!
>
> &nbsp;
> ### Numbered references:
>
> [1] Hausknecht, Matthew, and Peter Stone. "Deep recurrent Q-learning for partially observable MDPs." AAAI (2015).
>
> [2] Duan, Yan, et al. "RL2: Fast reinforcement learning via slow reinforcement learning." ICLR (2017).
>
> [3] Hafner, Danijar, et al. "Dream to control: Learning behaviors by latent imagination." ICLR (2020).
>
> [4] Hafner, Danijar, et al. "Learning latent dynamics for planning from pixels." *ICML*, 2019.
>
> [5] Doerr, Andreas, et al. "Probabilistic recurrent state-space models." *ICML*, 2018.
>
> [6] Buesing, Lars, et al. "Learning and querying fast generative models for reinforcement learning." Workshop on Prediction and Generative Modeling in Reinforcement Learning (2018).
>
> [7] Guez, Arthur, David Silver, and Peter Dayan. "Efficient Bayes-adaptive reinforcement learning using sample-based search." NeurIPS (2012).
>
> [8] Kaelbling, Leslie Pack, Michael L. Littman, and Anthony R. Cassandra. "Planning and acting in partially observable stochastic domains." Artificial intelligence, 1998

---

> > ### Author Response · Authors · 2023-11-22
> > **Follow Up**
> >
> > Dear Reviewer hsNK,
> >
> > As the rebuttal period is drawing to a close, we wanted to check whether we have addressed the concern you raised. We would also like to point out that we have added extensive new experimental results for new baselines. If you feel your concern (in regards to the Markovian latent space) has been addressed, we would appreciate it if you would consider raising your score.
> >
> > Thank you for your time,
> >
> > The authors

---

> > > ### Comment · Reviewer_hsNK · 2023-11-22
> > >
> > > Dear Authors,
> > >
> > > I thank you for your comments.
> > >
> > > **Implementation and learning a Markovian latent space**
> > >
> > > I am aware that stacking observations or encoding the history of observations is a well-established assumption in order to treat a POMDP as an MDP. However, my concern is very specific to the Dreamer model you adopted.
> > >
> > > DreamerV2's latent state is made of two components: one is deterministic and the other is stochastic. You can see the model in two ways: a non-Markovian model of stochastic states, where a summary of the previous stochastic states is accessed through the deterministic state, or a Markovian model, made of an hybrid deterministic-stochastic state.
> > >
> > > For WAKER, you claim to train an ensemble of transition models. However, it is not clear what component of the state you are predicting. In my view, there are three possibilities:
> > > * you are predicting both deterministic and stochastic states - this option would reflect the theory of the paper
> > > * you are focussing on the deterministic part of the state - this could be considered an approximation of the theory, as the Markovian assumption would hold but the deterministic state is only a proxy of the full state (you still have to feed it to the representation/prior model and sample in order to compute the stochastic part)
> > > * you are focussing on the stochastic part of the state - violates the Markovian assumption
> > >
> > > Having a look at Plan2Explore's implementation in the DreamerV2 repository, I would think that you are pursuing the second option. It would be useful if you could clarify this with a comment here and make sure this is clear in the paper as well, i.e. by stating what the transition model is predicting and why the state you are predicting is Markovian or an approximation of a Markovian state.
> > >
> > > ---
> > >
> > > I would consider raising my score, if you could address this one concern, given that the previous concerns have been addressed and that you added additional experiments to the paper, as requested by other reviewers.

---

> ### Author Response · Authors · 2023-11-22
> **Both the deterministic and stochastic state are predicted in our implementation**
>
> Thank you for getting back to us, and clarifying your concern! Our apologies that we misunderstood you in our original response.
>
> In our implementation, the ensemble is trained to predict **both the deterministic and the stochastic component of the next latent state**.
>
> To confirm that this is the case, you can check the code provided in the supplementary material of our original submission. In the file waker/configs.yaml, Line 77, the config variable "disag_target" (which represents the target value used to train the ensemble) is set to "feat". In waker/expl.py, Line 25, you can see that if config.disag_target is equal to "feat" then both the stochastic and deterministic parts of the state are used as targets to train the ensemble.
>
> As you have pointed out, this is a slight modification from the original Plan2Explore implementation, which only predicts part of the latent state. However, we agree that evaluating the uncertainty over the prediction of the entire latent state makes more sense in our context (for the reasons you have pointed out), which is why we implemented it this way.
>
> To make this clear in the paper, we have added a comment to the implementation details in the appendix under the paragraph "World Model Training" stating: *In our implementation of Plan2Explore, each member of the ensemble is trained to predict both the deterministic and stochastic part of the next latent state. This is slightly different to the Plan2Explore implementation in the official Dreamer-v2 codebase, where the ensemble only predicts part of the latent state. We made this modification because the derivation of our algorithm indicates that we should be concerned with the uncertainty over the prediction of the entire latent state*.
>
> We hope this clarifies your remaining concern!
>
> Kind regards,
>
> The authors

---

### Official Review · Reviewer_VCF1 · 2023-11-01

**Soundness:** 3 good
**Presentation:** 4 excellent
**Contribution:** 2 fair
**Rating:** 6
**Confidence:** 4

**Summary:**

This work considers the problem of pretraining a world model on reward-free data using a learned exploratory policy, and subsequently using the learned model to train task-specific policies (using a provided reward function) without any additional model learning. The key difference between this work and prior work is the strong focus on learning models that are *robust*: designing curricula for learning world models in the presence of multiple environment variations such that downstream policies learned using the model are more likely to generalize to unseen environment variations. The paper starts by treating the problem formally, then introduces a practical algorithm -- WAKER -- motivated by theoretical insights, and then compares the "robustness" of learned world models on OOD environment variations in a number of simulated environments.

**Strengths:**

- The paper is very well written, and provides sufficient background for readers unfamiliar with the problem to appreciate their theoretical result (Proposition 1). The theoretical treatment is not particularly surprising in itself, but helps motivate the proposed method which I appreciate. Assumptions are clearly stated (perfect model and latent policy exists), which seem pretty reasonable to make for the sake of analysis.
- The proposed method is intuitive and appears to be fairly straightforward to implement. I consider this a strength.
- Experiments are conducted on a reasonably large amount of tasks, which makes the results fairly trustworthy overall despite similar results for methods on many of the individual tasks. The proposed method achieves the most significant gains in the OOD evaluations, as one would expect based on the derivation of the algorithm.

**Weaknesses:**

I have three concerns at the moment:

- **Contributions.** While I really do consider simplicity a strength, the proposed method is undeniably incremental. At surface level, the main algorithmic contribution is automatic curricula by sampling of environments for which the current model error is large, which -- again, at surface level -- has been explored in numerous prior works as a means of guiding e.g. exploration, curricula for which rewards are available, gradient updates, or as model regularization (I will omit references to specific works here to avoid propagating my own biases, since I believe that the authors are aware of the literature, but can provide some during the discussion period if requested). This is in my opinion not a big problem given differences in problem setting as well as some of the more intricate algorithm details and theoretical contributions, but seems worth bringing up.
- **Experimental design.** It is evident from the empirical results that performance between WAKER and domain randomization is fairly similar for in-domain environment variations, and that most of benefit from the WAKER curricula is in the OOD test environments (Figure 4, Figure 13). However, this appears to mainly be because (1) the task variations considered have a natural order of difficulty (e.g. number of objects or terrain amplitude), (2) WAKER strongly favors sampling more difficult tasks (e.g. max allowed number of objects), and (3) the OOD variations considered here are formulated such that they are most similar to the task variations favored by WAKER (e.g. max possible number of training objects + 1). Therefore, it is not clear whether the models learned by WAKER really are robust or simply converge on a data distribution that is more favorable in these particular test environments.
- **Baselines.** The authors claim that there are no applicable baselines besides a naive domain randomization baseline that uniformly samples environment variations. However, there are several other manually designed curricula that would be very useful for validating the necessity for automatic curricula: (1) a baseline that strongly favors sampling the most difficult environments, (2) a baseline that is trained **only** on the most difficult environment, and (3) a baseline that performs automatic domain randomization by initially sampling from a small range of environments and gradually increases the range of sampled values (similar to ADR from https://arxiv.org/abs/1910.07113). Baselines 1 and 2 require some domain knowledge in that they need to know which environments are likely to be difficult (could be used as an oracle), but baseline 3 could easily be implemented without any domain knowledge by growing number of environments based on a linear schedule. It seems disingenuous to claim that there are no existing curriculum methods that would be applicable to this problem setting.

**Questions:**

I'd like the authors to address my questions and suggestions provided in the *weaknesses* section above. Addressing those comments are most likely to change my opinion. I have one additional clarification questions, though:
- The reward-free MDP introduced in Sec 2 is defined as a tuple that exclude the reward function R (as opposed to the standard MDP) but still includes a discount factor. What is the role of the discount factor in the reward-free setting, if any? Is this just for notational convenience in Sec 3?


Lastly, typo:
- P4, below Eq. 3: according the MDP

**Post-rebuttal:** I believe that the authors have addressed my biggest concerns, and I raise my score (5 -> 6) and confidence (3 -> 4) accordingly.

---

> ### Author Response · Authors · 2023-11-19
> **Response to Reviewer VCF1**
>
> Thank you for review! We will respond to each of the weaknesses you have raised below.
>
> &nbsp;
> ## Baselines
>
> Thank you for your suggestion for additional baselines! **We have added results for the additional baselines you have suggested - please see the general response above.**
>
> &nbsp;
> ## Contributions
>
> We agree that model uncertainty has been used extensively to guide exploration within a single MDP. Uncertainty in the value function is also often used to guide exploration in problems where the reward function is provided. Our main contribution is posing the problem of generating automatic curricula over a space of MDPs without rewards and proposing an algorithm for this problem. Our method actively explores the space of MDPs, in contrast to previous methods for reward-free exploration which do not allow for direct control (and therefore exploration) of the environment configurations during training. In this way, our method differs from these prior methods by directly generating curricula over the space of training environments.
>
> We discuss the relationship between our work and the aforementioned previous works in the related work section. Please let us know if you feel that there are additional existing works that we should discuss!
>
> &nbsp;
> ## Experimental Design
>
> We would like to point out that WAKER results in a statistically significant improvement over domain randomisation (DR) for the in-distribution evaluation results. The results in the table show that WAKER-M outperforms DR for almost every task. Therefore we disagree that similar performance is obtained compared to DR for the in-distribution evaluation.
>
> For the out-of-distribution (OOD) evaluation, we chose to evaluate difficult OOD environments as we think it is more important to evaluate how well the world model generalises to harder instances of the environments. The results for the new baselines show that the *Hardest Environment Oracle* baseline performs poorly for both the in-distribution evaluation, and the OOD evaluation on stairs terrain. **This demonstrates that simply mastering the most complex environment is insufficient to obtain strong performance on our evaluations.**
>
> &nbsp;
> ## Discount Factor
>
> Thank you for pointing out that the reward-free MDP does not require a discount factor. We have removed the discount factor from this definition, and added it to where we define MDPs with a reward function.

---

> > ### Comment · Reviewer_VCF1 · 2023-11-22
> > **Thank you**
> >
> > Thank you for the response and new experiments. I acknowledge that running new baselines during such a short time window is difficult, but it is clear that other reviewers also recognize the same need for baselines. I went over the new results, and I do believe that they have strengthened the paper. I do not have any further questions or major concerns, but expect the authors to provide the full baseline results in the event that the paper gets accepted. I lean towards acceptance and raise my score to reflect that.

---

> > > ### Author Response · Authors · 2023-11-22
> > > **Thank you for updating your review**
> > >
> > > Thank you for updating your review! We have just updated the paper to include results for the new baselines for three out of four domains. We will ensure that the full baseline results are complete in the final version.
> > >
> > > Kind regards,
> > >
> > > The authors

---

### Official Review · Reviewer_j9LV · 2023-11-04

**Soundness:** 3 good
**Presentation:** 2 fair
**Contribution:** 2 fair
**Rating:** 5
**Confidence:** 3

**Summary:**

This paper tackles the problem of learning world models through reward-free interactions that are robust to environments variations. First, it formulates the learning problem in terms of minimization of the minimax regret. Then, it proves that the problem formulated is equivalent to minimizing the expected error of the world model in the class of environments under an exploration policy. Hence, it designs an algorithm, called WAKER, to actively sample the environments on which the world model is trained on. Finally, it provides an empirical evaluation of the proposed approach in challenging domains, especially showing its superiority against naive domain randomisation.

**Strengths:**

- (Relevance) The paper tackles a very relevant problem, which is learning a robust model of the world from reward-free interactions with a class of environments;
- (Originality) The paper contributes some nice ideas, such as considering the model error in expectation under the worst-case policy.

**Weaknesses:**

- (Presentation) The first part of the paper seems to set a very ambitious goal, which is to tackle the general problem of learning a model of a class of environments that allows to approximately solve any RL task within the class. I would rather present a narrower scope of extending domain randomisation to reward-free settings through a model-based approach;
- (Strong assumptions) The set of assumptions does not look completely reasonable. Especially assuming the knowledge of a decoder that maps every environment to an MDP on latent states and a planning oracle for a setting that is essentially POMDP;
- (Empirical evaluation) Despite the very general problem formulation, the proposed algorithm is evaluated in very structured domains, where the variation between environments and tasks is fairly limited. Moreover, the set of baselines basically counts different variations of the proposed algorithm, without a real external competitor;
- (Limited theoretical contribution) The paper sells the theoretical characterization of the problem as an important contributions, but there is not much in terms of novel theoretical results.

This paper proposes an interesting model-based algorithm for domain randomisation in reward-free settings, which is arguably very relevant given the recent advancements in reward-free RL (in a single MDP) and robust RL/domain randomization. However, while the WAKER algorithm looks like a reasonable approach, from the provided empirical evaluation it is hard to say whether the procedure can actually work when facing more diverse classes of environments. Moreover, I think the presentation is not great for this work: The wording of the first sections seems to imply a substantial theoretical contribution, while the actual contribution is essentially an heuristic algorithm. For these reasons, I am currently providing a slightly negative evaluation. I am open to raise my score if the authors can convince me that the proposed algorithm can actually work with reasonable efficiency and the presentation can be reworked to narrow the scope to "model-based domain randomisation with active training". I report more detailed comments below.

(C1) The provided formulation of the problem is reasonable and clear, but the paper is overselling that contribution in my opinion. Everything that is said in the first sections is already known for a single environment (especially that the sub-optimality gap of a planning policy can be upper bounded with the approximation error on the transition model). The paper claims to extend that for multiple environments, but the extension become almost straightforward when the paper assumes the existence of a decoder from any of the environment to a single MDP on a latent state representation.

(C2) The regret definition in Eq. 1 is also reasonable, even though it is mostly a generalization gap rather than an online learning regret, but not really novel (e.g., Chen et al., Understanding domain randomization for sim-to-real transfer, 2022). The $\pi^\star_{\text{regret}}$ has also been formulated before under the name of "domain-randomization policy" (Chen et al., 2022) or "Bayes-optimal policy" (e.g., Ghavamzadeh et al., Bayesian reinforcement learning: A survey, 2015).

(C3) The paper essentially assumes a planning oracle for POMDPs (Assumption 1) as the original problem is a POMDP. This is not uncommon, but it is hardly reasonable, as the original problem is arguably computationally hard (Lusena et al., Nonapproximability results for partially observable Markov decision processes, 2001).

(C4) How can the representation model q be implemented? The requirement defined in Assumption 2, i.e., that a set of partially observable environments can be captured into a single MDP over a latent states, looks really strong.

(C5) The paper says the "aim is to train the world model such that the policies obtained are robust, as measured by minimax regret. However, we cannot directly evaluate regret as it requires knowing the true optimal performance." Is instead evaluating the regret against a world model trained for the test environment a reasonable "ideal" target?

(C6) The notion of policy is never defined, which makes unclear whether the paper refers to Markovian policies or general history-dependent policies.

**Questions:**

Can the authors address my comments above?

---

> ### Author Response · Authors · 2023-11-19
> **Response to Reviewer j9LV (1/2)**
>
> Thank you for your review! We have added results for **additional baselines** as described in the general response. We will respond to your individual comments below.
>
> &nbsp;
> > From the provided empirical evaluation it is hard to say whether the procedure can actually work when facing more diverse classes of environments.
>
> We would like to point out that our paper includes experimental results where we learn a **single world model** for **both the terrain and block pushing environments**. Due to space constraints, these results are in Appendix B.1. These experiments demonstrate that WAKER is effective when learning across very diverse domains.
>
> We have added further discussion of these results in the main paper to more strongly emphasise them.
>
> &nbsp;
> > (C1)  Theoretical contribution claims.
>
> Our work extends results for the sub-optimality gap in a single MDP with a known reward function to a **set of MDPs** with an **unknown reward function**. We have modified the wording in the introduction to make it clear that our theoretical contribution extends existing theoretical works. The contributions part of the introduction now reads:
>
> “*We extend existing theoretical results for MDPs* to prove that this problem is equivalent to minimising the maximum expected error of the world model across all environments under a suitable exploration policy”
>
> Extending these theoretical results to the setting of a space of MDPs with an unknown reward function enables us to analyse robustness (in terms of minimax regret) in the reward-free setting. Thus, this analysis serves to motivate our final practical algorithm (WAKER).
>
> &nbsp;
> > (C2) Problem Formulation
>
> The domain randomisation policy defined by Chen et al., 2022 optimises for the expected value in expectation across a distribution of MDPs. Likewise, the Bayes-optimal policy is a history-dependent policy that optimises the expected value in expectation across a prior distribution of MDPs. Because of the *expectation* over MDPs, these works optimise the *expected* regret.
>
> In our work, we address a different problem setting. We are interested in the *maximum regret* over a set of MDPs. As we reference in the paper (above Equation 1) this objective has been used to define a *robust* policy in a number of prior works (Dennis et al., 2020; Jiang et al., 2021a; Parker-Holder et al., 2022; Rigter et al., 2021). Intuitively, the minimax regret objective results in robust performance because to have low maximum regret, the policy must have a low sub-optimality gap across all environments.
>
> As noted above, domain randomisation (DR) optimises for the expected value assuming that all environments are equally likely. Therefore, DR does not optimise for minimax regret: the optimal DR policy may have very high regret in some environments if this improves overall performance in expectation. To optimise for minimax regret, we must consider the sub-optimality in the worst-case possible environment.
>
> &nbsp;
> > (C3) Assumption 1: Planning Oracle for POMDPs
>
> While we agree that we cannot expect to solve large POMDPs exactly, we think that assuming the optimal policy in the world model can be found is a suitable assumption for the sake of deriving and motivating our algorithm. Our opinion is echoed by Reviewer VCF1. As we write in the paper, our assumption is justified by the fact that we can generate *unlimited data* within the world model to optimise the policy, so there are no sample efficiency constraints.
>
> Recent works on world models (e.g. Hafner et al. 2021) have shown that optimising policies using large amounts of synthetic data generated by a world model obtains strong performance on difficult POMDPs (such as Atari).

---

> > ### Author Response · Authors · 2023-11-19
> > **Response to Reviewer j9LV (2/2)**
> >
> > > (C4) Assumption that a suitable latent representation can be learnt (Assumption 2)
> >
> > This assumption comprises two parts: First, that we can learn a Markovian latent space for a single POMDP, and second, that we can learn this latent space for a set of POMDPs.
> >
> > We begin by addressing the first part of this assumption. We consider a world model to be a model that utilises a recurrent neural network (RNN) to encode the history of previous observations into the latent representation (Lines 88 - 91). It is well-established in existing works on deep reinforcement learning that utilising an RNN enables non-Markovian problems (such as POMDPs or Meta RL) to be treated as Markovian [1, 2].
> >
> > In the context of world models specifically, this assumption is also common. For example, the seminal paper [3] addresses POMDPs (i.e. non-Markovian problems) with a recurrent world model. The authors of [3] state that “The latent dynamics define a Markov decision process (MDP; Sutton, 1991) that is fully observed because the compact model states are Markovian”. The same assumption is also made in other previous works on world models such as [4, 5, 6].
> >
> > The intuition for why this is a reasonable assumption is as follows. The RNN encodes the *history* of previous observations into the latent state. The history of observations is a sufficient statistic for the *belief* over partially observable variables [7]. It is well known that if we treat the belief as the state of an MDP, then the optimal behaviour in this *Markovian* model is the optimal behaviour for the original non-Markovian problem [8]. This idea is commonly referred to as the *belief MDP* [8].
> >
> > Thus, we have established that we can expect to learn a Markovian latent space for a single POMDP if the latent state contains memory of the history. To extend this to a *set* of POMDPs, we consider the following reasoning. Different POMDPs will generate different histories of states and observations. Therefore, the RNN will encode the histories for each POMDP into different latent states. Therefore, different POMDPs will be encoded to regions of the latent space.
> >
> > Our empirical results for training a single world model across both the terrain and block pushing environments in Appendix B.1 demonstrate that this approach can successfully learn powerful world models even when trained across diverse environments.
> >
> > &nbsp;
> > > (C5) Evaluating Robustness
> >
> > As discussed in the problem definition, we are interested in evaluating the *maximum regret* across a range of environments. To evaluate the *maximum* regret, knowing the optimal performance for a single test environment is not sufficient. We need to know the optimal performance for *every single possible environment*. This is infeasible, which is why we use the CVaR evaluation criterion as an alternative robustness measure in the paper for the results in Table 1.
> >
> > &nbsp;
> > > (C6) Policy Definition
> >
> > As we address POMDPs, we learn history-dependent policies. These policies are Markovian mappings from the recurrent latent state of the world model to a distribution over actions, and this corresponds to history-dependent policies in the real environment. We have added this definition to the paper (top of page 3).
> >
> > &nbsp;
> > &nbsp;
> > ### Numbered references:
> >
> > [1] Hausknecht, Matthew, and Peter Stone. "Deep recurrent Q-learning for partially observable MDPs." AAAI (2015).
> >
> > [2] Duan, Yan, et al. "RL2: Fast reinforcement learning via slow reinforcement learning." ICLR (2017).
> >
> > [3] Hafner, Danijar, et al. "Dream to control: Learning behaviors by latent imagination." ICLR (2020).
> >
> > [4] Hafner, Danijar, et al. "Learning latent dynamics for planning from pixels." *ICML*, 2019.
> >
> > [5] Doerr, Andreas, et al. "Probabilistic recurrent state-space models." *ICML*, 2018.
> >
> > [6] Buesing, Lars, et al. "Learning and querying fast generative models for reinforcement learning." Workshop on Prediction and Generative Modeling in Reinforcement Learning (2018).
> >
> > [7] Guez, Arthur, David Silver, and Peter Dayan. "Efficient Bayes-adaptive reinforcement learning using sample-based search." NeurIPS (2012).
> >
> > [8] Kaelbling, Leslie Pack, Michael L. Littman, and Anthony R. Cassandra. "Planning and acting in partially observable stochastic domains." Artificial intelligence, 1998

---

> ### Comment · Reviewer_j9LV · 2023-11-21
>
> I want to thank the authors for their detailed replies and their effort in extending the experimental results considering additional baselines.
>
> However, I am not fully satisfied over the answers to some of my comments, for which I am following-up below.
>
> > We would like to point out that our paper includes experimental results where we learn a single world model for both the terrain and block pushing environments. Due to space constraints, these results are in Appendix B.1. These experiments demonstrate that WAKER is effective when learning across very diverse domains.
>
> I have noticed this experiment while reviewing the paper, but found it hard to tell whether WAKER was doing good in that setting. This is way I think that providing a sense of the "ideal" performance (i.e., what we could achieve on the test task) is crucial.
>
> > As noted above, domain randomisation (DR) optimises for the expected value assuming that all environments are equally likely. Therefore, DR does not optimise for minimax regret: the optimal DR policy may have very high regret in some environments if this improves overall performance in expectation. To optimise for minimax regret, we must consider the sub-optimality in the worst-case possible environment.
>
> It is true that domain randomisation optimises for the expectation over the prior. However, Chen et al. (2022) exactly tackle the question of how large the minimax regret (they call it "gap") of the DR policy can be. Under their assumptions, it is sublinear.
>
> >  As we write in the paper, our assumption is justified by the fact that we can generate unlimited data within the world model to optimise the policy, so there are no sample efficiency constraints.
>
> What I was mentioning there is that even computing an optimal policy for a *known* POMDP is not tractable, irrespective of the number of available samples.
>
> > Thus, we have established that we can expect to learn a Markovian latent space for a single POMDP if the latent state contains memory of the history. To extend this to a set of POMDPs, we consider the following reasoning. Different POMDPs will generate different histories of states and observations. Therefore, the RNN will encode the histories for each POMDP into different latent states. Therefore, different POMDPs will be encoded to regions of the latent space.
>
> The authors are definitely right on this one. I shall rephrase the concern as "a suitable compact representation can be learned". Of course one can stack the belief MDPs of the different environments, but even computing the belief MDP for a *single* environment is intractable in general.
>
> > As discussed in the problem definition, we are interested in evaluating the maximum regret across a range of environments. To evaluate the maximum regret, knowing the optimal performance for a single test environment is not sufficient. We need to know the optimal performance for every single possible environment. This is infeasible, which is why we use the CVaR evaluation criterion as an alternative robustness measure in the paper for the results in Table 1.
>
> Perhaps one can reduce the number of sampled environments (say 10 instead of 100) to allow for this comparison. Otherwise, I still do not understand how we can evaluate the performance of WAKER against the ideal performance.

---

> ### Author Response · Authors · 2023-11-22
>
> Thank you for your response!
>
> > Chen et al. (2022)
>
> Thank you for clarifying the relevance of this paper. We now cite it along with other previous works that have considered maximum regret  (**Chen et al., 2022**; Dennis et al., 2020; Jiang et al., 2021a; Parker-Holder et al., 2022; Rigter
> et al., 2021). The core novelty of our work is that unlike these previous works we consider maximum regret in the *reward-free* setting.
>
> > Assumption 1 and tractability of POMDPs
>
> After Assumption 1, the paper now reads: *In practice, we cannot expect to find the exact optimal policy within the world model, however Assumption 1 enables an analysis of our problem setting.*
> We hope that you agree this will be sufficient to avoid any confusion surrounding this assumption for future readers. We, along with Reviewer VCF1, think this is a reasonable assumption for the sake of analysis.
>
> Many impactful works on deep RL (e.g. PPO) take some inspiration from theoretical motivation. However, the theoretical version of the algorithm cannot be implemented exactly in practice, and therefore the final algorithm is an approximation to the theoretically motivated version, which works well empirically. Thus, our approach of using some theoretical assumptions to later derive a more practical algorithm that works well empirically is a standard approach in deep RL research.
>
> > "a suitable compact representation can be learned"
>
> We believe that the fact that existing world model approaches obtain strong performance across many challenging POMDPs indicates that state-of-the-art world models are capable of learning a good approximation to the belief MDP.
>
> > Perhaps one can reduce the number of sampled environments (say 10 instead of 100)
>
> We do not think that only using 10 environments would be sufficient to cover the space of possible environments in our domains. Therefore, the proposed evaluation criteria would be highly sensitive to the 10 environments selected. Furthermore, as you have pointed out above, we cannot expect to solve any POMDP environment exactly. Therefore, we cannot exactly compute the ideal performance to compare against for any environment.
>
> The overarching goal of robust RL is to find policies that are *robust*, i.e. perform well across an entire range of possible scenarios. One possible definition for robustness is minimax regret. However, for the reasons discussed, performing an evaluation of the maximum regret is not possible. Therefore, to test robustness, we evaluate the performance on the worst 10 environments out of 100 (CVaR). A policy that is robust, and has low maximum regret, must perform well across all environments. Therefore, a robust policy must perform well in the worst 10% of environments. On the other hand, a policy that is not robust will not perform well in the worst 10% of environments. This is why the CVaR evaluation is a suitable metric in our problem setting.
>
> CVaR has been used as an evaluation metric in previous works on minimax regret robustness (Jiang, 2021). The out-of-distribution evaluations use standard expected value on the OOD environments as the evaluation metric.
>
> Jiang, Minqi, et al. "Replay-guided adversarial environment design." Advances in Neural Information Processing Systems 34 (2021): 1884-1897.
>
> ___
>
> We hope that this resolves your remaining concerns.
>
> Kind regards,
>
> The authors

---

> > ### Comment · Reviewer_j9LV · 2023-11-22
> >
> > I thank again the authors for their quick and detailed replies.
> >
> > I think I have now a clearer picture of what are the intended contributions of the paper. While I see some notable and interesting ideas, such as:
> > - extending domain randomisation to reward-free settings;
> > - the definition of the data-collecting policy for active learning of the world model (eq. 5);
> >
> > I also see that some of the concerns I expressed in my review are still valid, mostly:
> > - the baseline problem, which makes it hard to understand if WAKER is doing good or not (robustness is great, but not enough. One does not want a performance that is robustly bad);
> > - the theoretical contribution, which does not seem particularly strong given prior works and assumptions.
> >
> > For these reasons, I am still hesitant to say that the contribution is substantial enough for acceptance. I see that other reviewers are more positive, so I will discuss my concerns with them and the AC.
> >
> > Best wishes,
> >
> > Reviewer j9LV

---

> ### Author Response · Authors · 2023-11-22
> **Final clarification regarding evaluation**
>
> Thank you for being responsive during the rebuttal period, and for clarifying your current position.
>
> As a final clarification on our end, you have said that "one does not want performance that is robustly bad". Our results show that WAKER obtains stronger performance than the baselines in the worst 10% of cases (Table 1: Robustness results) as well as **similar or stronger performance on average** compared to the best baselines (Table 5: Average performance, in Appendix B.4), and stronger average performance on OOD environments (Table 2). Thus, our results show that WAKER obtains **improved robustness with similar or slightly improved average performance**. Therefore, these results show that WAKER does not obtain performance that is "robustly bad". If the performance was "robustly bad",  i.e. WAKER consistently obtains similar yet low rewards across all environments, then the average performance in Table 5 would decrease relative to the baselines.
>
> We hope this might help to address your remaining concern regarding the evaluation.
>
> Kind regards,
>
> The authors

---

### Author Response · Authors · 2023-11-19
**Global Response - Additional Baselines**

Thank you for your reviews! We have posted separate responses to each of them.

Following the suggestion of Reviewer VCF1, we have added results for the following **new baselines**:

- Re-weighting Oracle (RW-Oracle): Samples 80% of the time from the most complex environments, and 20% of the time from the domain randomisation distribution.
- Hardest Environment Oracle (HE-Oracle): Always samples the most complex instance of the environment.
- Gradual Expansion (GE): 20% of the time samples a new environment from the DR distribution, and 80% of the time samples randomly from previous seen environments.

Details about the baselines have been added to Appendix F. Note that **RW-Oracle and HE-Oracle require expert domain knowledge**, while GE does not.

Due to resource constraints, so far results have been added for Terrain Walker only (3 seeds per new baseline). ********************************************************************************************************************************************************************************We will add more complete results for the new baselines with more seeds near the end of the rebuttal period.******************************************************************************************************************************************************************************** We have moved most of the related work section to the appendix to make space to discuss the new results.

---

> ### Author Response · Authors · 2023-11-22
> **More results for additional baselines**
>
> Dear Reviewers,
>
> We have updated the paper so that it now **contains at least three seeds for all new baselines for three out of four domains**. In the final version of the paper, we will ensure that there are five seeds for all domains.
>
> We have also updated the discussion of the results in the paper. For your convenience, we will add a summary of that discussion here.
>
> &nbsp;
>
> ## New Results Summary
>
> The robustness results in Table 1 show that GE obtains very similar performance to DR. This is unsurprising, as GE does not bias sampling towards more difficult environments, and only modifies the DR distribution by gradually increasing the range of environments sampled. HE-Oracle obtains poor performance, demonstrating that focussing on the most challenging environment alone is insufficient to obtain a robust world model. This is expected from the analysis in Section 3.3 which shows that to obtain robustness we need the world model to have low error across all environments (not just the most complex one). For Terrain Walker and Terrain Hopper, RW-Oracle is a strong baseline, and obtains similar performance to WAKER. However, for Clean Up RW-Oracle obtains significantly worse performance than WAKER. This is likely because by focussing sampling environments with four blocks of any colour, RW-Oracle does not sample diverse enough environments to obtain good robustness. This demonstrates that even with domain knowledge, handcrafting a suitable curriculum is challenging.
>
> For the OOD environments, GE obtains similar performance to DR for the OOD environments. For the steep terrain OOD environments, HE-Oracle performs quite well as it focuses on sampling the steepest possible in-distribution terrain. However, HE-Oracle does not perform well on the stairs OOD environments, demonstrating that sampling a range of environments is necessary for strong OOD generalisation. RW-Oracle performs well on the OOD environments across all domains. However, RW-Oracle has the signiﬁcant draw-
> back that expert domain knowledge is required. These results demonstrate that by actively sampling more uncertain environments for exploration, WAKER leads to world models that are able to generalise more broadly to environments never seen during training, without requiring any expert domain knowledge.

---

### Meta-Review · Area_Chair_ihTp · 2023-12-07

**Metareview:**

This research addresses the challenge of training generally adaptable agents through reward-free exploration, emphasizing robust world models and minimizing regret. The proposed algorithm, WAKER, strategically selects environments for data collection based on estimated world model errors. It also explores unsupervised environment design and a task-agnostic setting, formulating the problem as a minimax regret objective. WAKER's effectiveness is demonstrated empirically, showcasing superior performance over domain randomization in terms of robustness and out-of-distribution settings. The work contributes valuable insights into reward-free learning, curricula design, and adaptive agent training.

This paper introduces a novel approach to learning robust world models by generating task-agnostic curricula. The motivation is well-founded, and the presentation is clear and didactic. Theoretical rigor is evident in the formalization of the problem within a minimax regret framework, with clear assumptions and proven regret bounds. The derivation to the proposed learning agent (WAKER) is cleverly executed. Experimental validation demonstrates WAKER's effectiveness in robust and out-of-distribution scenarios, providing practical insights.

On the other hand, the proposed work faces critical concerns regarding its assumptions and practical implementation. The assumption about the exploration policy's ability to cover the entire state-action distribution is questioned, particularly in complex tasks like locomotion with terrain variability. The methodology for learning task-specific policies raises concerns about potential suboptimality if the dataset lacks crucial state-action pairs. The feasibility of Assumption 1, meant to address this issue, is also questioned. Additionally, there are reservations about the chosen metric for prediction error and the method of collecting the test set, with recommendations for alternative metrics and improved sampling strategies. These concerns collectively highlight areas for refinement in the theoretical and practical aspects of the proposed work.

After reading the authors' rebuttals, the reviewers discussed their concerns, and even if some weak points still need to be properly addressed, we recommend this paper for acceptance, with the understanding that the authors will address the identified weaknesses and incorporate any necessary clarifications or improvements in the final revision. We are confident that the suggested changes will further enhance the paper's quality and contribution to the field.

**Justification For Why Not Higher Score:**

While the paper addresses a relevant problem, it may benefit from additional depth and a more extensive empirical validation.

**Justification For Why Not Lower Score:**

While some concerns have been identified, these are, in my view, manageable with targeted revisions.

---

### Decision · Program_Chairs · 2024-01-16

Accept (poster)